# Moving beyond post-hoc XAI: A perspective paper on lessons learned from dynamical climate modelling

Ryan J. O'Loughlin[1], Dan Li[2], Richard Neale[3], Travis A. O'Brien[4, 5]

[1]Philosophy Department, Queens College, City University of New York, New York, 11367, USA
[2]Department of Philosophy, Baruch College, City University of New York, New York, 10010, USA
[3]National Center for Atmospheric Research, Boulder, CO, 80305, USA
[4] Department of Earth and Atmospheric Sciences, Indiana University, Bloomington, 47405, USA
[5]Lawrence Berkeley Lab Climate and Ecosystem Sciences Division, Berkeley, CA 94720, USA

*Correspondence to*: Ryan J. O'Loughlin (ryan.oloughlin@qc.cuny.edu)

Moving beyond post-hoc XAI: A perspective paper on lessons learned from dynamical climate
2                    modeling

**Abstract.** AI models are criticized as being black boxes, potentially subjecting climate science to greater uncertainty.
Explainable artificial intelligence (XAI) has been proposed to probe AI models and increase trust. In this Review and
Perspective paper, we suggest that, in addition to using XAI methods, AI researchers in climate science can learn from
past successes in the development of physics-based dynamical climate models. Dynamical models are complex but
have gained trust because their successes and failures can sometimes be attributed to specific components or sub-
models, such as when model bias is explained by pointing to a particular parameterization. We propose three types of
understanding as a basis to evaluate trust in dynamical and AI models alike: (1) instrumental understanding, which is
obtained when a model has passed a functional test; (2) statistical understanding, obtained when researchers can make
sense of the modelling results using statistical techniques to identify input-output relationships; and (3) Component-
level understanding, which refers to modelers' ability to point to specific model components or parts in the model
architecture as the culprit for erratic model behaviors or as the crucial reason why the model functions well. We
demonstrate how component-level understanding has been sought and achieved via climate model intercomparison
projects over the past several decades. Such component-level understanding routinely leads to model improvements
and may also serve as a template for thinking about AI-driven climate science. Currently, XAI methods can help
explain the behaviors of AI models by focusing on the mapping between input and output, thereby increasing the
statistical understanding of AI models. Yet, to further increase our understanding of AI models, we will have to build
AI models that have interpretable components amenable to component-level understanding. We give recent examples
from the AI climate science literature to highlight some recent, albeit limited, successes in achieving component-level
understanding and thereby explaining model behavior. The merit of such interpretable AI models is that they serve as
a stronger basis for trust in climate modeling and, by extension, downstream uses of climate model data.

## 1. Introduction

Machine learning (ML) is becoming increasingly utilized in climate science for tasks ranging
from climate model emulation (Beucler et al. 2019), to downscaling (McGinnis et al. 2021),
forecasting (Ham, Kim, and Luo 2019), and analyzing complex and large datasets more
generally (for an overview of ML in climate science, see Reichstein et al. 2019; Molina et al.
2023; de Burgh-Day and Leeuwenburg 2023). Compared with physics-based methods, ML, once
trained, has a key advantage: orders of magnitude reduced computational expense. Along with
the advantages of ML come challenges such as assessing ML trustworthiness. For example,
scientists often do not understand why a neural net (NN) gives the output that it does because the
NN is a "black box."[1]
To build trust in ML, the field of explainable artificial intelligence (XAI) has become
increasingly prominent in climate science (Bommer et al. 2023). Sometimes referred to as
"opening the black box," XAI methods consist of additional models or algorithms intended to
shed light on why the ML model gives the output that it does. For example, Labe and Barnes
(2021) use an XAI method, layer-wise relevance propagation, and find that their NN heavily
relies on datapoints from the North Atlantic, Southern Ocean, and Southeast Asia to make its
predictions.
While XAI methods can produce useful information about ML model behaviors, these methods
also face problems and have been subjected to critique. As Barnes et al. (2022) note, XAI
methods "do not explain the actual decision-making process of the network" (p. 1). Additionally,
different XAI methods applied to the same ML model prediction have been shown to exhibit
discordance, i.e., yielding different and even incompatible "explanations" for the same ML
model (Mamalakis et al. 2022). Discordance in XAI is not unique to climate science. Krishna et
al. (2022) find that 84% of their interviewees (ML practitioners across fields who use XAI
methods) report experiencing discordance in their day-to-day workflow and when it comes to
resolving discordance, 86% of their online user study responses indicate that ML practitioners
either employed arbitrary heuristics (e.g., choosing a favorite method or result) or simply did not
know what to do.
As Molina et al. (2023) note, "identifying potential failure modes of XAI, and uncertainty
quantification pertaining to different types of XAI methods, are both crucial to establish
confidence levels in XAI output and determine whether ML predictions are 'right for the right
reasons'" (p. 8). Rudin (2019) argues that, instead of attempting to use XAI to explain ML
models post hoc, scientists ought to build interpretable models informed by domain expertise
from the outset. Speaking about explainability in particular, Rudin writes, "many of the [XAI]
methods that claim to produce *explanations* instead compute useful summary statistics of
predictions made by the original model. Rather than producing explanations that are faithful to
the original model, they show trends in how predictions are related to the features" of the model
input (2019, p. 208).
Regardless, XAI methods will likely continue to be widely applied due to ease of use and as
benchmark metrics for XAI methods are proposed and implemented (Hedström et al. 2023;
Bommer et al. 2023). In some cases, XAI methods are applied with great success, e.g.,
(Mamalakis et al. 2022) found that the input x gradient method fit their ground truth model with
a high degree of accuracy. However, we believe that more progress can be made in establishing

---

[1] Note that computer scientists have proposed various conceptual approaches to articulate "transparency" (e.g., Lipton 2016). However, we aim to offer conceptual clarity for ML applications specifically in climate science by comparing different types of understanding of ML and dynamical climate models.

trust in ML-driven climate science, especially as an increasing number of researchers start
incorporating ML into climate research.
In this Review and Perspective paper, we target readers with expertise in traditional approaches
for climate science (e.g., development, evaluation, and application of traditional Earth System
Models) who are starting to utilize ML in their research and who may see XAI as a tempting way
to gain insight into model behavior and to build confidence. In this perspective, we draw from
some ideas in philosophy of science to recommend that such researchers leverage the expanding
array of freely available ML learning resources to move beyond post hoc XAI methods and aim
for *component-level* understanding of ML models. By "component" we mean a functional unit of
the model's architecture, such as a layer or layers in a neural net. By "understanding" we mean
knowledge that could serve as a basis for an explanation about the model. We distinguish
between three levels of understanding:
**Instrumental understanding:** knowing *that* the model performed well (or not); e.g.,
knowing its error rate on a given test.
**Statistical understanding:** being able to offer a reason why we should trust a given ML
model by appealing to input-output mappings. These mappings can be retrieved by
statistical techniques.
**Component-level understanding:** being able to point to specific model components or
parts in the model architecture as the cause of erratic model behaviors or as the crucial
reason why the model functions well.
These levels concern the degree to which complex models are intelligible or graspable to
scientists (De Regt and Dieks 2005; Regt 2017; Knüsel and Baumberger 2020). Therefore, our
proposal has a narrower but deeper focus than recent philosophy of science accounts of
understanding climate phenomena *with* or *by using* ML and dynamical climate models (Knüsel
and Baumberger 2020; Jebeile, Lam, and Räz 2021). We are concerned with understanding,
diagnosing, and improving model behavior to inform model development.
Instrumental understanding, while clearly necessary, is fairly straightforward and is a
prerequisite for any explanation of model behavior. It involves knowing the degree to which a
model fits some data (Lloyd 2010; Baumberger et al. 2017). It may also involve knowing
whether the model both fits some data *and* agrees with simpler models about a prediction of
interest or whether the model has performed well on an out-of-sample test (e.g., Hausfather et al.
2020) or according to other metrics (e.g., Gleckler et al. 2008).
However, in this Review and Perspective paper, we will only focus on the other two types of
understanding. Statistical understanding can be gained via traditional XAI methods but does not
require knowledge of the model's innerworkings, i.e., its components and/or architecture (see
Sect. 2 below). In contrast, component-level understanding *does* involve knowledge of the
model's innerworkings. Therefore, component-level understanding allows scientists to offer
causal explanations that attribute ML model behaviors to its components. Scientists need to build
and analyze their models in such a way that they can understand how distinct model components
contribute to the model's overall predictive successes or failures rather than merely probe model
data to yield input-output mappings. The latter is emblematic of traditional XAI methods.
Our recommendation to strive for component-level understanding is inspired by how dynamical
climate models have been built, tested, and improved, such as those in the coupled model
intercomparison projects (CMIP). Therefore, a novel contribution of this paper is in the linking
of existing climate model development practices to practices that could be employed in ML
model development.
In CMIP, when models agree on a particular result, scientists sometimes infer that the governing
equations and prescribed forcings shared by the models are responsible for the models' similar
results. As Baumberger et al. (2017) put it, "robustness of model results (combined with their
empirical accuracy) is often seen as making it likely, or at least increasing our confidence, that
the processes that determine these results are encapsulated sufficiently well in the models" (p.
11; see also Hegerl et al. 2007; Kravitz et al. 2013; Lloyd 2015; Schmidt and Sherwood 2015;
O'Loughlin 2021). Conversely, when climate models exhibit biases or errors, scientists can often
point to specific parameterizations or sub-models as the likely cause (e.g., Gleckler et al. 1995;
Pitari et al. 2014; Gettelman et al. 2019; Zelinka et al. 2020); O'Loughlin 2023), although
models can get the right answer for the wrong reasons (e.g., see Knutti 2008).
To be clear, there are limits to how much component-level understanding can be achieved in
CMIP. Dynamical climate models exhibit fuzzy (rather than sharp) modularity, meaning that the
behavior of a fully coupled model is "the complex result of the interaction of the modules—not
the interaction of the results of the modules" (Lenhard and Winsberg 2010, p. 256). Climate
scientists are familiar with a related problem: the difficulty in explaining how climate models
generate (or not) emergent phenomena like the Madden Julian Oscillation (Lin et al. 2024).
Despite these difficulties, philosophers and other scholars of climate science have documented
successes in attributing model behavior to individual model components in the climate science
literature (Frigg et al. 2015; Carrier and Lenhard 2019; Touzé-Peiffer et al. 2020; Pincus et
al.2016; Hall and Qu 2006; Hourdin et al. 2013; Notz et al. 2013; Oreopoulos et al. 2012;
Mayernik 2021; Gettelman et al. 2019; O'Loughlin 2023). These successes do not imply
anything like a "full" or "complete" understanding of all model behavior, rather, the component-
level understanding of climate model behavior comes in degrees (Jebeile et al. 2021).
Fortunately, we see component-level understanding exemplified in ML-driven climate science to
some extent already (Beucler et al. 2019; Kashinath et al. 2021; Bonev et al. 2023, see Sect. 4
below). Indeed, the thinking behind physics-informed machine learning, which incorporates
known physical relations into the models from the outset (Kashinath et al. 2021;Wang et al.
2022; Cuomo et al. 2022), often involves component-level understanding. Thus, our proposal is
an endorsement of these ongoing best practices, a recognition of the relationship between the
evaluation of dynamical models and data-driven models, and a warning about the limits of
statistical understanding. In addition, there is a concurrent need to establish the trustworthiness
of ML models as ML-driven climate science potentially becomes increasingly used to inform
decision makers (NSF AI Institute for Research on Trustworthy AI in Weather, Climate, and
Coastal Oceanography (AI2ES)). While decision makers themselves do not need to understand
exactly how a model arrives at the answer it does, they may desire an explanation of the model's
behavior that comes from a credible expert. One way to establish credibility is to be able to
explain ML model behavior by appealing to the innerworkings of the model, which requires
component-level understanding of the model. In this way, component-level understanding can
serve as a basis for trust in ML-driven climate science.
The remainder of the paper is structured as follows. In Sect. 2, we give an overview of XAI in
climate science and explain the idea of statistical understanding and how XAI can only give us
statistical understanding. In Sect. 3, we detail the notion of component-level understanding and
demonstrate it using examples from CMIP. In Sect. 4, we show how component-level
understanding is achievable in ML. In Sect. 5, we conclude and make suggestions for ML-driven
climate science, including describing some resources that interested readers might utilize to build
the expertise in ML model design necessary to probe, build, and adapt models in a way that is
amenable to component-level understanding.


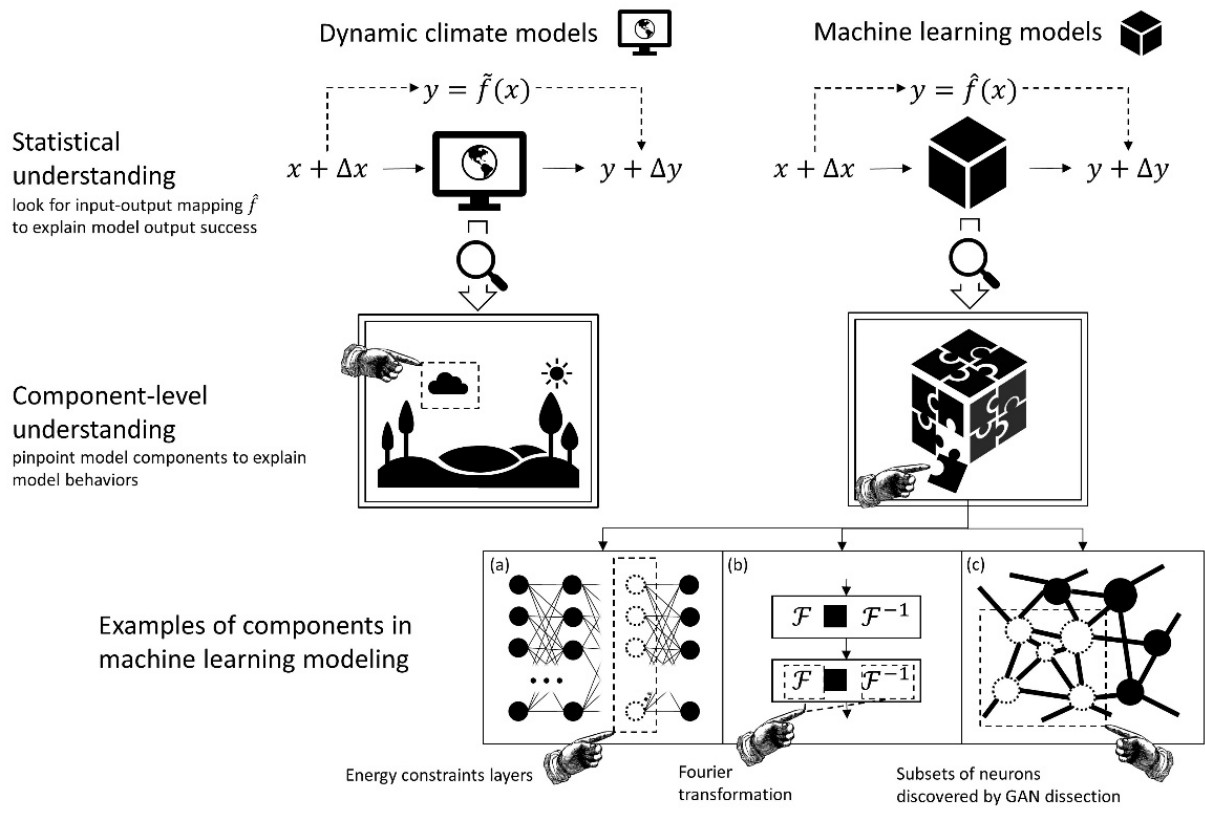


*Figure 1. Scientists can obtain statistical understanding of models by seeking input-output mapping, e.g., via perturbation experiments. To acquire component-level understanding, one needs to be able to pinpoint specific components to explain models' erratic behaviors or successes. This has been done in dynamic climate modeling, e.g., by pointing to cloud parameterization as a means to improve modeling outcomes. We offer three examples of component-level understanding in machine learning. In panel (a), Beucler et al. (2021) design layers of neurons in their neural network to enforce energy conservation and improved model outcome. In panel (b), Bonev et al. (2023) use spherical Fourier transformation to ensure Fourier Neural Operators perform with climate data. In panel (c), Bau et al. (2019) use a method called GAN dissection to identify which subsets of neurons control parts of images that correspond to semantics (e.g., trees or doors).*


## 2. Post-hoc XAI in climate science and statistical understanding


XAI methods are intended to shed light on the behavior of complex opaque ML models. As
Mamalakis et al. (2022b) put it, XAI "methods aim at a post hoc attribution of the NN prediction
to specific features in the input domain (usually referred to as attribution/relevance heatmaps),
thus identifying relationships between the input and the output that may be interpreted physically
by the scientists" (p. 316). XAI methods are typically applied to ML models which are multi-
layer, convolutional, recurrent neural networks, and/or tree ensembles.
The general idea behind XAI methods is to attribute the predictive success of the model's output
(i.e., the model's prediction or decision) to subsets of its input in supervised ML. Broadly, there
are two conceptual approaches to achieve this.[2] One approach is to figure out how the changes in
input affect the output. For example, Local Interpretable Model-agonistic Explanation (LIME)
first perturbs an input data point to create surrogate data near said data point. Then, after the
trained ML model classifies the surrogate data, LIME fits a linear regression using classified
surrogate data and measures how model output can be attributed to features of the surrogate data
manifold. In this way, LIME attributes the predictive success for the actual data point to a subset
of input features. Note that L stands for "local" because LIME starts with perturbing specific
classificatory instances rather than with global classification.
Another commonly used method is Shapley Additive explanation (SHAP), which is based on
calculating the Shapley values of each input feature. Shapley values are cooperative game
theoretic measures that distribute gains or costs to members of a coalition. Roughly put, Shapley
values are calculated by repeatedly randomly removing a member from the group to form a new
coalition and calculating the consequent gains and then averaging all marginal contributions to
all possible coalitions. In the XAI context, input features will have different Shapley values,
denoting their different contribution to the model's predictive success. E.g., see (Chakraborty et
al. 2021; Felsche and Ludwig 2021; Cilli et al. 2022; Clare et al. 2022; Grundner et al. 2022; W.
Li et al. 2022; Xue et al. 2022)
Another approach relies on treating a trained black box model as a function to understand how
the input-output mapping relationship is represented by this function. For example, vanilla
gradient (also known as saliency) is an XAI method that relies on calculating the gradient of
probabilities of output being in each possible category with respect to its input and
backpropagates the information to its input. In this way, vanilla gradient quantifies the relative
importance of each element of the input vector with respect to the output, thereby attributing the
predictive success to subsets of input. E.g., see Balmaceda-Huarte et al. 2023; Liu et al. 2023; He
et al. 2024.[3]
Let's examine how XAI methods yield statistical understanding in a detailed example. González-
Abad et al. (2023) use the saliency method to examine input-output mappings in three different
convolutional neural nets (CNNs) which were trained and used to downscale climate data. They
computed and produced accumulated saliency maps which account for "the overall importance
of the different elements" of the input data for the model's prediction (p. 8). One of their results
is that, in one of the CNNs, air temperature (at 500hPa, 700 hPa, 850hPa, and 1000 hPA)
accumulates the highest relevance for predicting North American near-surface air temperature,

---

[2] Yuan et al. (2023) break down the various XAI methods into four categories. They divide those related to manipulating input-output into perturbation-based methods and surrogate-based methods (e.g., LIME). They divide the methods that rely on model parameter values into gradient-based methods (e.g., gradient) and decomposition-based method (e.g., layerwise relevance propagation ).

[3] Yet another commonly used XAI method, layerwise relevance propagation, computes how each neuron contributes to other neurons' activations, thereby highlighting the subsets of the input that dominantly contribute to the output. E.g., see (Gordon, Barnes, and Hurrell 2021; Toms, Barnes, and Hurrell 2021; Labe and Barnes 2021; 2022a; 2022b; Rader et al. 2022; Diffenbaugh and Barnes 2023).

although different regions are apparently more relevant than others to the models' predictions
(see their figure 6, p. 12). In other words, it appeared that the CNN had correctly picked up on a
relationship between coarse resolution temperature at certain geopotential heights on the one
hand, and higher resolution near-surface air temperatures on the other hand.
In this way, XAI methods yield information that can be helpful for making a model's results
intelligible. E.g., it puts a scientist in the position to say, "this model was picking up on aspects
A, B, and C of the input data. These aspects contributed to prediction X, a prediction that seems
plausible." This exemplifies what we call "statistical understanding", i.e., being able to offer a
reason why we should trust a given ML model by appealing to statistical mappings between
input and output. Statistical techniques are often used to obtain these mappings by relating
variations in input to variations in output. Post hoc XAI methods can typically yield this type of
understanding. Note that this is not the same as explaining the innerworkings of the model itself,
or what we call "component-level understanding," because the explanation does not attribute the
model behaviors to ML model components, but rather is focused on input-output mapping.
While XAI methods can give statistical understanding of model behaviors, this type of
understanding has limitations. The general limitation is a familiar one, i.e., that "while XAI can
reveal correlations between input features and outputs, the statistics adage states: 'correlation
does not imply causation'" (Molina et al. 2023, p. 8)[4]. Even if genuine causal relationships
between input and output can be established, we still do not know how the ML model produces a
certain output. To answer this question, ideally, we would like to know the causal role played by
(at least) some of the components making up the model. We would like to know about at least
some processes, mechanisms, constraints, or structural dependencies inside of the model, rather
than merely probing the ML-model-as-black-box post hoc, from the outside. While XAI methods
can yield information that seems plausible and physically meaningful, this information may be
irrelevant with respect to how the model actually arrived at a given decision or prediction (Rudin
2019; Baron 2023). This, in turn, can undermine our trust in the model for future applications. In
contrast, with component level understanding, the causal knowledge is more secure and can also
inform future development and improvement of the model in question and ML models in
general.
**3. Understanding and Intelligibility in CMIP**
Dynamical models are complex but have gained trust because their successes and failures can
sometimes be attributed to specific components or sub-models, such as when model bias is
explained by pointing to a particular parameterization. Indeed, the practice of diagnosing model
errors pre-dates the Atmospheric Model Intercomparison Project (AMIP; Gates 1992). For
example, differences in the representation both of radiative processes and of atmospheric

---

[4] To be more precise, we interpret this quote as saying that correlation does not (logically) entail causation.
Correlation may be a sign that there is a causal relation in play, and correlations between events often lead us to try
and relate events causally.

stratification at the poles were featured in an evaluation of why 1-D models diverged from a
GCM in their estimate of climate sensitivity (see Schneider 1975).
Later, in one of the diagnostic subprojects following AMIP, Gleckler et al. (1995) attributed
incorrect calculations of ocean heat transport to the models' representations of cloud radiative
effects. They first found that the models' implied ocean heat transport was partially in the wrong
direction—northward in the Southern Hemisphere. They inferred that cloud radiative effects
were the culprit, explicitly noting that atmospheric GCMs at the time of their writing were
"known to disagree considerably in their simulations of the effects of clouds on the Earth's
radiation budget (Cess et al. 1989), and hence the effects of simulated cloud-radiation
interactions on the implied meridional energy transports [were] immediately suspect" (Gleckler
et al. 1995, p. 793). They recalculated ocean heat transport using a hybrid of model data and
observational data. When they did this, they fixed the error—ocean heat transport turned
poleward. The observational data used to fix the error were of cloud radiative effects. In other
words, they substituted the output data linked to the problematic cloud parameterizations (a
*component* of the models) with observational data of cloud radiative effects. This substitution
resulted in a better fit with observations of and physical background knowledge of ocean heat
transport.
One may argue that substituting model components merely exemplifies statistical understanding
because it concerns the input and output data of the models, which, in Glecker et al.'s case, are
cloud-radiation and ocean heat transport. Yet, this would be misguided. Gleckler et al. isolated
the cloud components as the causal culprit behind why the models produced biased ocean heat
transport data. There is also a physically intelligible link between cloud radiative forcing and
ocean surface heat, so the diagnosis made scientific sense. In this way, scientists can diagnose
and fix climate models.
Many more recent cases of error diagnosis also aim to identify problematic parameterizations
(e.g., see (Hall and Qu 2006; O'Brien et al. 2013; Pitari et al. 2014; Bukovsky et al. 2017;
Gettelman et al. 2019; but see Neelin et al. 2023 for current challenges). In CMIP6 in particular,
there is an increased focus on process-level analysis (Eyring et al. 2019; Maloney et al. 2019). In
process-level analysis, scientists examine bias in the simulation of particular processes which
are, in turn, linked to one or more parameterizations, i.e., components within a whole GCM.[5]
Moreover, CMIP-endorsed model intercomparison projects (MIPs) also center on particular
processes or parameterizations, such as the cloud feedbacks MIP (Webb et al. 2017) and the land
surface, snow and soil moisture MIP (van den Hurk et al. 2016).[6]

[5] Note that while processes and model components are linked, neither is reducible to the other. E.g., a coupler is a
component in a GCM but it is not a real-world climate process; conversely, there is no cloud feedback
parameterization but cloud feedbacks are a real-world climate process.
[6] These examples are in stark contrast to the pessimism about understanding climate models that some philosophers
of science have emphasized (Lenhard and Winsberg 2010) and others have rebutted (Frigg, Thompson, and Werndl
2015; Carrier and Lenhard 2019; Touzé-Peiffer, Barberousse, and Treut 2020; O'Loughlin 2023; Easterbrook 2023).


The practice of updating model parameterizations during model development also demonstrates
an interest (and success) in achieving component-level understanding. We provide two examples
here: one associated with the radiative transfer parameterization in the Community Atmosphere
Model and another associated with the physical representation of stratocumulus clouds in
boundary layer parameterizations. With respect to the radiative transfer component
(parameterization), Collins et al. (2002) noted that, at the time their paper was written, studies
had "demonstrated that the longwave cooling rates and thermodynamic state simulated by GCMs
are sensitive to the treatment of water vapor line strengths." Collins et al. used this knowledge—
along with updated information about absorption and emission of thermal radiation by water
vapor—to update the radiation parameterization in the Community Atmosphere Model. This
component-level improvement led to substantial improvements in the models' simulated climate.

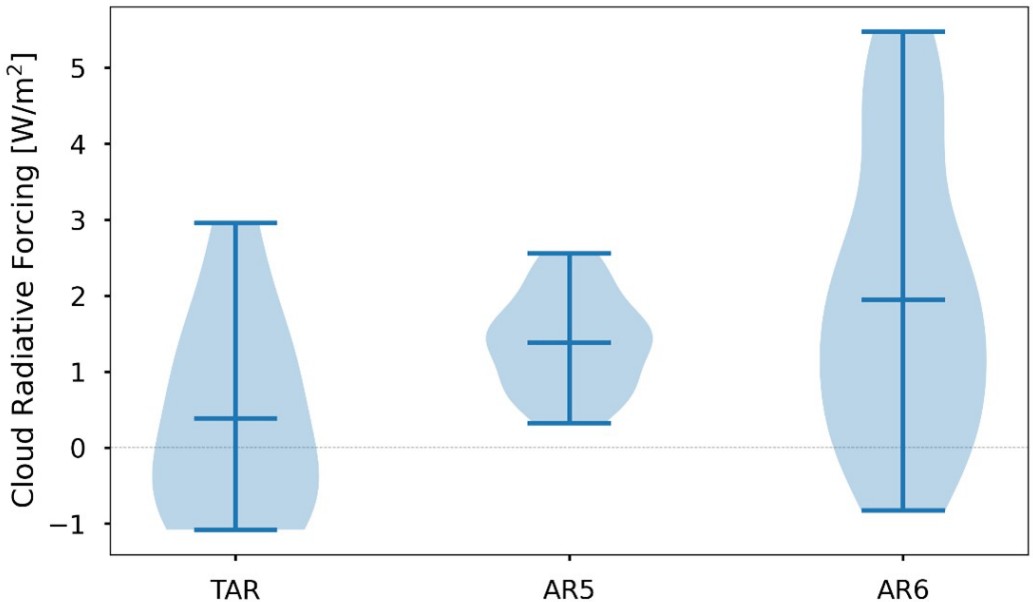


*Figure 2. Changes in the distribution of estimated cloud radiative forcing (CRF) across three generations of IPCC Assessment Reports: 3 (TAR, published in 2001), 5 (AR5, 2014), and 6 (AR6, 2021). AR4 is omitted because data necessary to estimate CRF are not readily available. Estimates of simulated CRF were acquired by manual digitization of Figure 7.2 of Stocker et al. (2011) and by multiplying the equilibrium climate sensitivity and cloud feedback columns from Tables S1 and S2 of Zelinka et al. (2020). As the distribution of estimated cloud radiative forcing shifts upwards from TAR to AR5 to AR6, the figure shows that in AR5 and AR6, cloud feedbacks are largely positive. Indeed, AR6 states with high confidence that "future changes in clouds will, overall, cause additional warming" (Forster et al., 2021, p. 1022), yet it was not clear in TAR whether cloud feedbacks were positive. The increasing confidence in positive cloud feedbacks is partially due to improved boundry-layer parameterization, which demonstrates modelers' component-level understanding.*


Regarding stratocumulus cloud parameterization in climate models, targeted developments
following the Third Intergovernmental Panel on Climate Change (IPCC) Assessment Report
reduced uncertainty in estimates of cloud feedbacks to the extent that the 6[th] IPCC Assessment

Report now states with high confidence that "future changes in clouds will, overall, cause additional warming" (p. 1022). This systematic change in cloud radiative forcing is demonstrated in Figure 2. It was not clear in the Third IPCC Assessment Report (TAR) whether cloud feedbacks were positive or negative, and the TAR noted in particular that the "difficulty in simulation of observed boundary layer cloud properties is a clear testimony of the still inadequate representation of boundary-layer processes" (TAR 2001), p. 273). Around this time, researchers developed improved boundary layer parameterizations with the goal of improving the representation of low, boundary layer clouds. For instance, Grenier and Bretherton built on a standard 1.5-order boundary layer turbulence parameterization in which turbulent mixing is treated as a diffusive process related to the amount of turbulent kinetic energy (TKE) and in which TKE is treated as a conservative, prognostic quantity. Their key additions to the 1.5-order turbulence approach were (1) a more accurate numerical treatment of diffusion in the vicinity of step-function-like jumps in temperature and humidity (inversions) and (2) contribution of cloud-top radiative cooling to the production of TKE. These two ingredients allow the turbulence parameterization to emulate the physics that drive stratocumulus clouds. Variations on the parameterization of (Grenier and Bretherton (2001) and other similarly sophisticated boundary layer parameterizations have been included in numerous weather and climate models, leading to improvements in the simulation of stratocumulus clouds specifically and general improvements in model climatology.

In certain circumstances component-level responsibility for particular model biases can be determined. As an example, the Community Earth System Model 2 (CESM2) was recognized as exhibiting a too-large climate sensitivity—one that did not appear in standard CMIP simulations. This behavior was discovered in a surprising way. Zhu et al. 2021 had shown an instability in the simulation of the last glacial maximum, a much colder period than present day, using CESM2. This instability did not exist in CESM. By reverting to the original, component-level microphysics scheme the model behaved as expected, and erroneous specification of microphysical particle concentrations were discovered and remedied. More generally, the understanding and observational constraint of ice microphysics is a challenge as demonstrated by the very large variations in ice water path across CMIP models. Using Perturbed Parameter Estimation (PPE, e.g., Eidhammer et al. 2024) can also reveal component level sensitivities and shortcomings.

We take the above cases from CMIP to indicate that climate scientists aim for component-level understanding of their models, which relates to a standard that climate models be at least somewhat *intelligible*. Adopting the idea of "intelligibility" from philosopher of science de Regt (2017) we can say that a complex model is intelligible for scientists if they can recognize qualitatively characteristic consequences of the model without performing exact calculations. Intelligibility is facilitated by having models made up of components. In dynamical models, these components typically represent real-world processes, even in cases of empirically based parameterizations. More generally, knowing that a model component plays a particular role—

either representing the process as designed or a role later discovered during model
development—in a climate simulation is invaluable for reasoning about the behavior, successes,
and biases of the GCM as a whole.
The climate modeling community has long strived for component-level understanding and
intelligibility. This is especially evident in the work on climate model hierarchies, i.e., a group of
models which spans a range of complexity and comprehensiveness Jeevanjee et al. (2017).
Writing nearly two decades ago, Issac Held (2005) identified model hierarchies as necessary if
we wish to understand both the climate system and complex climate models:
we need a model hierarchy on which to base our understanding, describing how the dynamics
change as key sources of complexity are added or subtracted... (p. 1609)
…the construction of such hierarchies must, I believe, be a central goal of climate theory in
the twenty-first century. There are no alternatives if we want to understand the climate
system and our comprehensive climate models. Our understanding will be embedded within
these hierarchies. (p. 1610)

Along similar lines, and before the advent of CMIP, Stephen Schneider (1979) wrote that
…the field of climate modeling needs to "fill in the blanks" at each level in the hierarchy of
climate models. For only when the effect of adding one change at a time in models of
different complexity can be studied, will we have any real hope of understanding cause and
effect in the climatic system. (p. 748)

These appeals to climate model hierarchies highlight how component-level understanding is a
longstanding goal in climate modeling (see also Katzav and Parker 2015). This is not to say that
component-level understanding automatically translates to understanding all model behaviors.
Emergent properties such as equilibrium climate sensitivity may elude explanation. Even when
components such as cloud parameterizations are appealed to as causally relevant for higher ECS
values (e.g., Zelinka et al. 2020), it must be granted that these cloud parameterizations *interact*
*with* other components and pieces of the overall GCM. That is, GCMs exhibit fuzzy
modularity—sub-model behaviors do not add up linearly or in an easy-to-understand way
(Lenhard and Winsberg 2010). So there may be a more complete explanation detailing how, as a
whole, the GCM simulates a higher ECS. Producing a complete explanation may prove elusive,
however, to the extent that GCMs are *epistemically opaque* or have such a high degree of
complexity that human minds cannot track all of the relevant information (Humphreys 2009).[7]
Therefore, we do not regard our three proposed types of understanding as exhaustive—perhaps a

---

[7] This complexity includes both the impossibility of fully knowing a climate model's code in its entirety, and the impossibility of being able to follow the calculations as the model steps forward in time. With today's GCMs, humans can do neither of these things.

component-interaction or structural type of understanding ought to be theorized and strived for
as well.
However, the examples from earlier in this section show how the goal of component-level
understanding is regularly achieved, overall model complexity notwithstanding. Having achieved
such understanding, scientists can be more confident that their models have indeed captured
some truths about the target systems, and they are thereby justified to increase their confidence in
these complex models. In the climate modeling literature, component-level understanding
routinely leads to model improvements.
We end this section with a brief discussion distinguishing between component-level and
statistical understanding. Overall, our analysis is in the same spirit as that of Knüsel and
Baumberger (2020) who argue that data-driven models and dynamical models alike can be
understood through manipulating the model so that modelers can qualitatively anticipate model
behaviors. However, not all manipulations are equal. Manipulating input data and seeing
associated changes in output data does not tell you how the model produces its output. The
hierarchy of understanding we propose—instrumental, statistical, and component-level—
concerns the degree to which and ways in which a model is intelligible or graspable (Knüsel and
Baumberger 2020; Jebeile, Lam, and Räz 2021). Complex models are intelligible or graspable
just in case, and to the degree that, their behavior can be qualitatively anticipated or explained
(De Regt and Dieks 2005; Lenhard 2006). From our perspective, component-level understanding
puts scientists into a position to better anticipate and better explain model behavior. In general,
statistical understanding can help us answer questions such as "do the input-output relations of
the model make sense and, if so, in what way do they make sense?" This is great for finding out
whether the model's behavior is consistent with expectations across a variety of cases. This may
also involve manipulating input and examining associated changes in output, to better anticipate
future model behavior (Knüsel and Baumberger 2020; Jebeile, Lam, and Räz 2021). However,
this is distinct from learning *why* the model behaves the way it does. To answer this distinct
question, we need to know how the model is working, which, in turn, involves knowing
something about the pieces making up the model. Hence, component-level understanding is
called for. This is exactly the type of understanding that we see aimed for, and often grasped, in
CMIP experiments.
Component-level understanding often involves a different kind of knowledge related to model
architecture and beyond input-output relationships. On the one hand it can demonstrate that you
know what role the component is playing in the model—this shows some knowledge of model-
building. It may also be helpful for answering a wider range of what-if-things-had-been-different
questions. Finally, and potentially the clearest benefit of component-level understanding, is that
it can tell one what needs to be fixed in cases of error. This should produce additional trust in the
modeling enterprise more generally.[8]
**4. Lessons learned: examples of component level understanding in ML**
Component-level understanding is not the privilege solely of dynamic climate modeling. ML
models can be built with intelligible components as well, although their components look very
different from those in dynamic models. In this section, we offer three examples in which ML
researchers are able to acquire component-level understanding of model behaviors by
intentionally designing or discovering model components that are interpretable and intelligible.
*4.1 Attributing model success with physics-informed machine learning*
Our first example involves physics-informed machine learning, i.e., machine learning
incorporated with domain knowledge and physical principles (Kashinath et al. 2021). Model
success can be attributed to a specific component in a neural net, if it is known that said
component in the neural net is performing a physically relevant role for a given modeling task.
Beucler et al. (2019; 2021) augment a neural net's architecture via layers which enforce
conservation laws that are important for emulating convection (see Figure 1, panel a). These laws
include enthalpy conservation, column-integrated water conservation, and both long- and short-
wave radiation conservation. The conservation laws are enforced "to machine precision"
(Beucler et al. 2021). Following Beucler et al. (2019) and because this neural net has a physics-
informed *architecture*, we will use the acronym NNA. NNA is trained on aqua-planet simulation
data from the Super-Parameterized Community Atmosphere Model 3.0. NNA's results are
compared with those of two other neural nets: one *unconstrained* by physics (NNU) and another
"softly" constrained through a penalization term in the *loss* function (NNL; see Beucler et al.
(2019) for further discussion).
All three NNs are evaluated based on the mean squared errors (MSE) of their predictions and
based on whether their output violates physics conservation laws (they call this a P-score). While
NNU has the highest performance in a baseline climate—i.e., a climate well-represented by the
training data—NNA and NNL each outperform NNU in a 4k warmer climate (see Beucler et al.
2019, Table 1), which is impressive since generalizing into warmer climate is particularly
challenging for ML models (Rasp et al. 2018; Li 2023). These results may indicate that NNU
performed better in the baseline climate for the "wrong" reasons. Indeed, NNU had a far lower
P-score in both the baseline and the 4k warmer climate cases.
Beucler et al. (2021) further show that NNA predicts the total thermodynamic tendency in the
enthalpy conservation equation more accurately than the other NNs—"by an amount closely

---

[8] This is not to say that component-level understanding is necessarily superior to statistical understanding. E.g.,
knowing about a robustly detected statistical relationship could be more valuable than knowing how a single model
component functions, especially since many important model behaviors arise from interactions between multiple
model components.

related to how much each NN violates enthalpy conservation" (p. 5). The particular layer in
NNA responsible for enthalpy conservation is obviously the explanation for this result. This case
therefore exemplifies component-level understanding, which was straightforward because of
Beucler et al.'s choice of model design.
It should be noted that both NNA and NNL perform well in the 4k warmer climate and, more
generally, "[e]nforcing constraints, whether in the architecture or the loss function, can
systematically reduce the error of variables that appear in the constraints" (Beucler et al. 2021, p.
5). This suggests that, when thinking purely about model performance, physical constraints do
not necessarily need to be implemented *in* the model's architecture. However, compared with
NNL, Beucler et al.'s use of NNA facilitates straightforward component-level understanding.
The component-level understanding is straightforward because we know that, by virtue of the
physics knowledge built into the model's architecture, NNA obeys conservation laws as it is
trained and as it is tested. We can draw an analogy with dynamical climate models. NNL is to
NNA as bias-corrected GCM simulations are to ones which capture relevant physical processes
with high-fidelity to begin with. Knowing that a model produces a physically consistent answer
for physical reasons is a stronger basis for trust than merely knowing that a model produces
physically consistent answers due to post-hoc bias correction.

*4.2 Explaining model error in a case of Fourier Neural Operators*
Another example involves a recent development in using machine learning to solve partial
differential equations: the Fourier neural operator (FNO) pioneered by Li et al. (2021). The
innovation of FNO is the application of Fourier transforms to enable CNN-based layers that learn
"solution operators" to partial differential equations in a scale-invariant way. Building on Li et al.
(2021), Pathak et al. (2022) demonstrated that training an FNO network on output from a
numerical weather prediction (NWP) model produced a machine learning model that emulates
NWP models with high fidelity and efficiency. A key challenge noted by Pathak et al. (2022),
however, was a numerical instability that limited application of the FNO model to forecasts of
lengths less than 10 days.
Analysis of the instability ultimately led the group to hypothesize that the instability was due to a
specific component of the FNO model: the Fourier transform itself. The problem they identified
was that the sine/cosine functions employed in Fourier transforms are the eigenfunctions of the
Laplace operator on a doubly-periodic, Euclidean geometry, whereas the desired problem (i.e.,
NWP) is intrinsic to an approximately spherical geometry. In essence, the Earth's poles represent
a singularity that Fourier transforms on a latitude-longitude grid are not well-equipped to handle.
Bonev et al. (2023) adapt the FNO approach to spherical geometry by utilizing spherical
harmonic transforms with the Laplace-operator eigenfunctions for spherical geometries as basis
functions, in lieu of Fourier transforms. These eigenfunctions, the spherical harmonic functions,
smoothly handle the poles as a natural part of their formulation. Bonev et al. (2023) report that
the application of spherical harmonic transforms, rather than Fourier transforms, results in a
model that is numerically stable up to at least O(100) days and possibly longer.
The application of spherical transformations stabilizes the FNO model. Bonev et al. were able to
fix the FNO because they could pinpoint the Fourier transformations, a component of the FNO
model, demonstrating scientists' component-level understanding.[9]
*4.3 GAN dissect for future applications in ML-driven climate science*
The final example comes from generative adversarial networks (GANs) in computer vision. Bau
et al. (2018) identify particular units (i.e., sets of neurons and/or layers) in a neural net as
causally relevant to the generation of particular classes within images such as doors on churches.
They demonstrate that these units *are* actually causally relevant by showing what happens when
said units are ablated (essentially setting them to 0).
The example demonstrates component-level understanding because the units in question are
manipulated. Components within the architecture of the model are turned on and off and the
resultant effects are observed.[10] This puts us in a position to say, for example, "these neurons are
responsible for generating images of trees, and we know this because turning more of these
neurons on yields an image with more trees (or bigger trees) and vice versa. Moreover, the other
aspects of the image are unchanged no matter what we do to these neurons." Bau et al. (2018)
also show that visual artefacts are causally linked to particular units and can be removed using
this causal knowledge.
This case is analogous to the study from Gleckler et al. (1995) as described in Sect. 3 above.
Recall that the cloud radiative effects from the GCMs were "turned off" (substituted out and
replaced with observational data) and the calculations of ocean heat transport improved.
Scientists can make sense of model error because they know that a certainty deficiency in GCMs,
at the time, involved components of the GCMs responsible for representing clouds. In the same
way, Bau et al. (2018) are able to intervene on generations of images by linking units in their
model to particular types of image classes and examining what happens to the overall image
when these units are manipulated. Note that this is distinct from the closely related method of
ablating specific subsets of input data, which is more closely aligned with XAI and can therefore
yield statistical understanding (e.g., see Brenowitz et al. 2020; Park et al. 2022).
While GAN dissect isn't typically used in climate science research, GANs are beginning to be
adopted for some climate applications (Besombes et al. 2021; Beroche 2021). Additionally, there
are potential future applications such as in atmospheric river detection Mahesh et al. (2023). In

---

[9] Fourier transformations turn out to be useful in other contexts of ML-driven climate science because scientists can use them to understand neural networks behaviors as combinations of filters, e.g., (Subel et al. 2023).

[10] As a reminder to the reader, by "component" we mean a functional unit of the model's architecture, which includes the "units" described by (Bau et al. 2018).

any case, this example demonstrates yet again how component-level understanding is achievable
with ML.

**5. Discussion/Recommendations for practice**
In this Review and Perspective paper we have argued that component-level understanding ought
to be strived for in ML-driven climate science. The value of component-level understanding is
especially evident in the FNO problem described previously (Sect. 4.2 above). Instrumental
understanding allowed the group to identify a performance issue (numerical 'issues' in the polar
regions) that led to numerical instability. While the group did not employ any XAI—statistical
understanding—approaches, it is clear that they would have been of limited value in identifying
the underlying cause of the numerical instability, since XAI methods only probe input-output
mappings. Ultimately the problem was identified and later solved by applying component-level
understanding of the FNO network: knowledge that a component of the network implicitly (and
incorrectly) assumed a Euclidean geometry for a problem on a spherical domain.
However, a potential objection is that component-level understanding is unnecessary because
XAI methods can simply be evaluated against benchmark metrics. For example, Bommer et al.
(2023) propose five metrics to assess XAI methods, focusing especially on the methods' output
data (referred to as "explanations"). These include:
**Robustness** of the explanation given small perturbations to input
**Faithfulness,** by comparing the predictions of perturbed input and those of unperturbed input
to determine if a feature deemed important by the XAI method does in fact change the
network prediction

**Randomization**, which measures how the explanation changes by perturbing the network
weights, similar to the robustness metric, the thinking is that "the explanation of an input x
should change if the model changes or if a different class is explained" (Bommer et al.
(2023), p.8)

**Localization**, which measures agreement between the explanation and a user-defined region
of interest
**Complexity,** a measure of how concise the highlighted features in an explanation are, and
assumes that "that an explanation should consist of a few strong features" to aid
interpretability (Bommer et al. 2023, p. 10).
Insofar as the metrics are deemed desirable, we agree that such an approach could help establish
trust in XAI. However, we view such benchmarks as complementary to, rather than a substitute
for, component-level understanding. This is because benchmarks yield a sort of second-order

statistical understanding. That is, such metrics are largely focused on aspects of input and output data produced by a given XAI method. They are, in a sense, an XXAI method, an input-output mapping to help make sense of another input-output mapping.

Therefore, our recommendation is that ML-driven climate science strive for component-level understanding. This will aid in evaluating the credibility of model results, in diagnosing model error, and in model development. The clearest path to component-level understanding in ML-driven climate science would likely involve climate scientists building, or helping build, the ML models that are used for their research and implementing physics-based and other background knowledge to whatever extent feasible (Kashinath et al. 2021; Cuomo et al. 2022). Clear standards could also be developed for documenting ML architecture, training procedures, and past analyses, including error diagnoses (O'Loughlin 2023). Perhaps a model intercomparison project could be developed to systematically evaluate ML behavior across diverse groups of researchers. Lastly, with component-level understanding as a goal to strive for, scientists can better develop hybrid models where both ML and dynamic modeling components are employed.

An increasing range of free or low-cost, high-quality resources are now available to enable researchers who are not (yet) experts in ML to gain a deep and practical level of understanding of modern ML model designs and applications. Some examples of free, high-quality resources include:

- Practical Deep Learning for Coders - 1: Getting started (fast.ai)
  - Related: GitHub - fastai/fastbook: The fastai book, published as Jupyter Notebooks
- Introduction - Hugging Face NLP Course
- How Diffusion Models Work - DeepLearning.AI

Back in 2005, Held wrote that climate modeling "must proceed more systematically toward the creation of a hierarchy of lasting value, providing a solid framework within which our understanding of the climate system, and that of future generations, is embedded" (p. 1614). We think there is a parallel need in ML-driven climate science, i.e., to develop systematic standards for the use and evaluation of ML models that aid in our understanding of the climate system. Striving for component-level understanding of ML models is one way to help achieve this.

**Code/data availability: No data was used or generated for this research**

**Author contributions:** DL conceptualized the project with assistance of RO and TO; RO wrote and prepared the manuscript with writing contributions from DL and TO; DL conceptualized and created the key visualization (figure 1); TO conceptualized and created figure 2; RN contributed to the writing and revision of the text

**Funding support:** This research was supported in part by (a) the Environmental Resilience Institute, funded by Indiana University's Prepared for Environmental Change Grand Challenge initiative; (b) the Andrew Mellon Foundation; (c) a PSC-CUNY Award, jointly funded by The Professional Staff Congress and The City University of

New York; (d) the U.S. Department of Energy, Office of Science, Office of Biological and Environmental Research,
Climate and Environmental Sciences Division, Regional & Global Model Analysis Program under Contract Number
DE-AC02-05CH11231 and under award Number DE-SC0023519.

**Competing interests:** At least one of the (co-)authors is a member of the editorial board of Geoscientific Model
Development.

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
