# Peer review of "Moving beyond post-hoc XAI: A perspective paper on lessons learned from dynamical climate modelling"

_EGUsphere, 2023_

## Referee Comment (RC1)

The paper is clear, well-organized, and discusses a very interesting approach, XAI methods, that could help scientists overcome epistemic opacity of ML-based models. I believe this paper can contribute in the reflection within the climate science community on the use of machine learning for modelling on the one side, and, on the other side, in the philosophy of science debates on understanding through climate models and epistemic opacity of machine learning techniques. Indeed, the two original contributions of the paper are, first, to use philosophical concepts in order to analyse the possible difficulty in the use of ML-based models in climate science, and, second, to discuss promising novel methods, i.e. XAI methods. But a number of revisions are needed. In what follows, I give some suggestions.

For the philosophy of science part:

As an intersdiciplinary researcher working with climate scientists, I find important to not introduce new terms that actually refer to already existing concepts in philosophy, but also to make connection with the relevant philosophy of science literature. Yet, there is now a rich discussion in philosophy of science on understanding through climate models and epistemic opacity of machine learning techniques which would be worth being cited and used for this paper. More precisely, I think that the following aspects of the paper should be revised:

1_ The authors argue that what they call "component-level understanding" should and can be reached with climate models but also with ML-based models with the help of XAI methods. They also argue that CMIP is a place where component-level understanding has successfully increased.

1.1_ However, this understanding — that seems similar to what Frisch (2015) calls "analytical understanding" — comes with the assumption that climate models are modular and that the interactions between the different modules (or model components) can be grasped and anticipated. But this modularity has been qualified as "fuzzy" by Lenhard and Winsberg (2010) and therefore scientists are facing what Lenhard and Winsberg call "entrenchment". Clearly this is in conflict with what the authors are claiming in this paper. That is why I believe the authors should engage with this debate (and revise the paper accordingly, all along the paper, not only at the beginning of the paper). I don't think that it would undermine their argument at all but will make it more nuanced and stronger; what they call "component-level understanding" might still be an ideal to pursue in the scientific practice. I also recommend the authors to read and cite the paper on modularity by Lenhard (2018).

1.2_The paper of Lenhard and Winsberg (2010) also demonstrates the failure of CMIP in making intercomparisons and thereby reaching what the authors call "component-level understanding". In this draft, the scientific references used to support the claim that AMIP / CMIP allowed for more component-level understanding are not recent (e.g. first paragraph p. 5 for instance or Glecker et al. 1995 cited p. 8), thus it would be nice that the authors explore whether it is the aim of CMIP6/7 using recent examples/illustrations. There might be another interesting paper on this topic, the paper of Touzé-Peiffer, Barberousse and Le Treut (2020).

1.3_ Another well-discussed issue in philosophy of science that makes "component-level understanding" difficulty to reach in the case of climate models is the epistemic opacity of climate models simulations, that models be dynamical or ML-based. Here are examples of such papers: Knüsel and Baumberger 2020; Kawamleh 2021; Jebeile, Lam and Räz 2021.

2_ The authors put forward three kinds of understanding, instrumental understanding, statistical understanding and component-level understanding.

2.1_ One would expect this taxonomy to be connected to what philosophers have already said about understanding with models, or to be motivated by what scientists tell about their own practices. Thus, in (Knüsel and Baumberger 2020) and (Jebeile, Lam and Räz 2021), the authors put forward different dimensions of understanding with models that therefore comes in degree. In particular, notably following the work of de Regt and Dieks cited in the paper, Jebeile, Lam and Räz (2021) use these evaluative criteria of understanding with models: intelligibility, representational accuracy, empirical accuracy, physical consistency, delimiting the domain of validity. Is this explicitation of "understanding with models" useful for this paper? For example, the

difference between "statistical understanding" and "component-level understanding" is that only the latter meets intelligibility, no? (I am just curious here, this might not be crucial for the paper though).

2.2_ What about the concept of "process understanding" used by climate scientists? It is usually referring to the aim of fundamental research. Is it not covered by the concept of "component-level understanding"?

2.3_ In the paper, what is the role of this taxonomy after all? Couldn't the authors simply introduce the definition of "component-level understanding" (or process understanding) and argue that it can be reached in ML-based modeling with the help of XAI methods (where we could imagine that only statistical understanding is reached)?

2.4_ It would also be interesting to have a characterization of this taxonomy: is instrumental understanding a weaker form of understanding than statistical understanding? Is statistical understanding in turn a weaker form than component-level understanding?
Or do they overlap?

2.5_ In the hierarchy of models envisioned by Held (2005), is he referring to model component? In the quotations given p. 11, he instead speaks about the dynamics.
Another paper on hierarchy of climate models is (Katzav and Parker 2015).

Regarding the contribution of this paper for the scientific practice:

1_ It would be worth defining what "computational efficiency" of machine learning is (introduction p. 2) as it is usually the main motivation in the use of machine learning. It would be important for this paper to clarify what it means.

2_ It seems that the authors are assuming (or have to assume) that there is some kind of isomorphism between layers in neural net and model components. Can they clarify their position on this? (cf. second paragraph p. 4).

3_ Can it not be that search for "component-level understanding" is actually search for "representational accuracy". Trying to correct for previous idealizations and parameterizations seem to be line with the "natural" direction of scientific research, no? (This is what is assumed in Jebeile and Roussos 2023; Baldissera Pacchetti, Jebeile and Thompson 2024).

4_ As it is, section 4.3 fails to be persuasive because GAN does not to apply to climate science. The authors should explain whey they believe that, in the future, GAN will be applied to ML-driven climate science.

Minor comments:

1_ In what sense, do AI models entail "greater uncertainty"? Could you specify what you mean: is that that the outputs / predictions of models are more uncertain? (abstract, p. 2)

2_ What is actually the "functional test" that a model has to pass in order to provide instrumental understanding? (abstract, p. 2)

3_ Some technical terms should be (better) defined: "layer-wise relevance propagation"; "attribution/relevance heatmaps"; "multi-layer, convolutional recurrent neural networks", "tree ensembles" (p. 6); distinction between "specific classificatory instances" and "global classification" (p. 7); "P-score" (p. 13)

4_ There should be no bracket after "Gettelman et al. 2019." (p. 5).

5_ References are needed to support the claim that "In addition, there is a concurrent need to establish the trustworthiness of ML models as driven climate science potentially becomes increasingly used to inform decision makers" (p. 5).

6_ In the introduction of section 4, the authors write "we offer three examples in which ML researchers are able to acquire component-level understanding of model behaviors by intentionally designing or discovering model components that are interpretable and intelligible." This sentence seems to suggest that "interpretable and intelligible" model components will bring component-level understanding (p. 13). Could the authors clarify this point?

7_ The authors should write what the acronym PDEs is referring to in all letters (p. 14)

8_ The authors should indicate the year of Pathak et. Al (p. 14) and add the reference in the list of references.

References indicated in this review:

Frisch, M. 2015. Predictivism and Old Evidence: a Critical Look at Climate Model Tuning. *European Journal for Philosophy of Science* 5 (2):171–190.

Lenhard, J., and Winsberg, E. 2010. "Holism, Entrenchment, and the Future of Climate Model Pluralism". *Studies in History and Philosophy of Science* Part B, 41(3):253–262.

Lenhard, J. (2018). Holism, or the erosion of modularity: a methodological challenge for validation. *Philosophy of Science, 85*(5) 832–844.

Jebeile, J., Lam, V. & Räz, T.  (2021) Understanding climate change with statistical downscaling and machine learning. *Synthese* 199, 1877–1897. https://doi.org/10.1007/s11229-020-02865-z

Kawamleh, S. (2021). Can machines learn how clouds work? The epistemic implications of machine learning methods in climate science. *Philosophy of Science*, *88*(5).

Knüsel, B., & Baumberger, C. (2020). Understanding climate phenomena with data-driven models. *Studies in History and Philosophy of Science Part A*. https://doi.org/10.1016/j.shpsa.2020.08.003.

Baldissera Pacchetti, M., J. Jebeile, and E. Thompson, 2024: For a Pluralism of Climate Modelling Strategies. *Bull. Amer. Meteor. Soc.*, https://doi.org/10.1175/BAMS-D-23-0169.1, in press.

Touzé-Peiffer L, Barberousse A, Le Treut H. The Coupled Model Intercomparison Project: History, uses, and structural effects on climate research. *WIREs Clim Change*. 2020; 11:e648. https://doi.org/10.1002/wcc.648

Katzav, J., & Parker, W. S. (2015). The future of climate modeling. Climatic Change, 132, 475–487.

---

## Community Comment (CC1)

This paper "**Moving beyond post-hoc XAI: Lessons learned from dynamical climate modeling**" (or a variation of it) is vitally important to understanding how AI/ML techniques will be adopted by the climate science community. Since climate science differs from other scientific disciplines in the impossibility of performing controlled laboratory experiments at scale, any claims need to stand the test of time or through extensive cross-validation against historical results. That's the primary reason that new ideas are treated with suspicion in climate science (or in any of the other earth sciences) -- without experimental validation, all new models or hypotheses appear equally in doubt. Doesn't matter if they are AI-generated or by humans, climate scientists will file it away with all the other candidates, likely to be ignored as they can't easily be validated, e.g. as a new semiconductor device model can be validated by a lab experiment.

That is a daunting challenge but if nothing else, ML provides an extensive selection of cross-validation approaches that climate science can borrow from. That needs to be stated up front. As ML can easily generate matches to virtually any kind of data due to the magic of non-linear neural networks, cross-validation is necessary to weed out the many that over-fit the observations. The AI literature is full of cross-validation citations, as that is the lifeblood metric of the discipline of machine learning. The majority of neural network model fits would fail on non-training data without the benefit of rigorous cross-validation testing.

Yet, applying cross-validation alone is not enough. Climate science is further complicated by the fact that there is no consensus physics model that explains a climate behavior as erratic as the El Nino Southern Oscillation (ENSO). A cross-validation of an ML experiment matching ENSO observations would also need to explore possibly novel physical mechanisms, especially with respect to the fluid dynamics, that the GCM simulations may not be considering. That is part of the original promise of AI – that of discovering *emergent behavior* not previously considered. So that needs to be stated as well, as it could turn out that dynamical climate modeling could learn from XAI, as it's not outside the realm of possibility that an AI experiment could find something that a GCM formulation missed.

In the context of XAI, it's also important to acknowledge an oft-overlooked AI approach that links pure mathematical physics modeling to that of a physics-unaware neural network – that of *symbolic regression* coupled with a genetic algorithm to explore the solution space. Whereas a neural network will generate a tangled web of nonlinear interactions that are difficult to reverse engineer to a possible physical mechanism, a symbolic regression application will apply algebraic/calculus formulations with a selection of inputs to optimize a fit and perform cross-validation. Since the formulation is presented symbolically, it takes far less effort for a human to discern and sort through possible plausible physical mechanisms leading to the discovered equation.

There are certainly drawbacks to applying symbolic regression in comparison to a neural net, but there are cases that could be cited for offering a promising approach. One in particular is the independent discovery made by a symbolic regression tool called **Eureqa** in the plausible explanation of the mechanism behind the quasi-biennial oscillation (QBO) of equatorial stratospheric winds. Eureqa was able to discover a nonlinear interaction coupled to differential equation that reasonably fit to the data over the span of years that QBO data as collected, 1952 to the present. The human effort necessary was to supply a physical mechanism for the equation and numerical parameters. This was reported as an exact match to a non-linear lunar tidal interaction with the annual cycle, as described in *Mathematical Geoenergy*, P. Pukite, D. Coyne, D. Challou (Wiley/AGU, 2019). Alas, this AI-adjacent model has not

gained any traction in the climate science community as it faces the same challenges of acceptance as any other model outside of the consensus.

So concerning the paper, in the discussion section, qualities such as robustness, faithfulness, *etc* are suggested. In practice, the critical factors include the 3 P's of plausibility, predictivity, and parsimony. Plausibility covers the allowable physics. Predictivity covers how well the model matches the observations, by minimizing the error. Parsimony covers the simplicity/complexity of the model in terms of the number of degrees of freedom (DOF) or terms expressed.  Any standard model optimization technique features a *Pareto front* characterization curve that tracks predictivity (error minimized) versus parsimony (complexity minimized) as a metric. All symbolic reasoning tools feature this as a optimization metric.

[Figure]

A more rudimentary example of a Pareto front optimized symbolic reasoning solution is also described in Mathematical Geoenergy. This uses several factors to model the global temperature with a simple arithmetic superposition. The Pareto front is shown in the lower-right below.

[Figure]

**Figure 17.4** ☼ Symbolic reasoning solution to global temperature series showing a composite of main factors along a Pareto front of complexity and accuracy.

Suggestion then is to incorporate symbolic regression in addition to the neural network approach. Each of these approaches can be more suitable for different types of problems. Symbolic regression with genetic algorithms offers a more interpretable model which could be preferable in scientific applications where understanding the underlying phenomena is crucial – closer to what XAI implies. Neural networks might be more suitable for problems involving high-dimensional data, which also describes the requirements of a complex climate system. The jury is still out what will eventually work, perhaps a combination of the two, but should state the possible choices and lessons learned.

---

## Author Comment (AC1)

Reviewer comments and our responses

[Responses in **blue; line numbers refer to tracked-changes revision document unless otherwise indicated**]

1. Reviewer 1 comments

The paper is clear, well-organized, and discusses a very interesting approach, XAI methods, that could help scientists overcome epistemic opacity of ML-based models. I believe this paper can contribute in the reflection within the climate science community on the use of machine learning for modelling on the one side, and, on the other side, in the philosophy of science debates on understanding through climate models and epistemic opacity of machine learning techniques.
Indeed, the two original contributions of the paper are, first, to use philosophical concepts in order to analyse the possible difficulty in the use of ML-based models in climate science, and, second, to discuss promising novel methods, i.e. XAI methods. But a number of revisions are needed. In what follows, I give some suggestions.

For the philosophy of science part:

As an intersdiciplinary researcher working with climate scientists, I find important to not introduce new terms that actually refer to already existing concepts in philosophy, but also to make connection with the relevant philosophy of science literature. Yet, there is now a rich discussion in philosophy of science on understanding through climate models and epistemic opacity of machine learning techniques which would be worth being cited and used for this paper. More precisely, I think that the following aspects of the paper should be revised:1_ The authors argue that what they call "component-level understanding" should and can be reached with climate models but also with ML-based models with the help of XAI methods. They also argue that CMIP is a place where component-level understanding has successfully increased.

1.1_ However, this understanding — that seems similar to what Frisch (2015) calls "analytical understanding" — comes with the assumption that climate models are modular and that the interactions between the different modules (or model components) can be grasped and anticipated. But this modularity has been qualified as "fuzzy" by Lenhard and Winsberg (2010) and therefore scientists are facing what Lenhard and Winsberg call "entrenchment". Clearly this is in conflict with what the authors are claiming in this paper. That is why I believe the authors should engage with this debate (and revise the paper accordingly, all along the paper, not only at the beginning of the paper). I don't think that it would undermine their argument at all but will make it more nuanced and stronger; what they call "component-level understanding" might still be an ideal to pursue in the scientific practice. I also recommend the authors to read and cite the paper on modularity by Lenhard (2018).

**Thank you for this comment. We have added a discussion (see footnote 6, lines 368-375) in the manuscript. We have added citations throughout. The notion of "component-level" understanding is indeed similar to Lenhard and Winsberg's (2010) idea of "analytic understanding" that Frisch (2015) also discusses. One of us has argued in print (O'Loughlin 2023) that much of Lenhard and Winsberg's argument is flawed because they misrepresent how climate models are built and**

their empirical evidence is inadequate to show what they want it to show. Lenhard and Winsberg define "analytic understanding" as "the ability to tease apart the various sources of success and failure of a simulation and to attribute them to particular model assumptions" and they argue "Unfortunately, analytic understanding is hard or even impossible to achieve... One *cannot* trace back the effects of assumptions because the tracks get covered" (emphasis added). If they are right, then model improvement would seem like a miracle! Indeed, O'Loughlin (2023) lists several examples where climate modelers successfully attribute model error to particular components.  However, we agree that fuzzy modularity limits how much a model's behavior can be understood. We have revised the manuscript accordingly. In our revisions, (e.g., lines 138-140) we also clarify that component-level understanding comes in degrees.

More generally, several philosophers and other scholars of climate modeling (e.g., Baumberger et al., 2017; Carrier & Lenhard, 2019; Frigg et al., 2015; Touzé-Peiffer et al., 2020; Easterbrook 2023) have also responded to Lenhard and Winsberg. An illustrative recent example comes from Steve Easterbrook's 2023 book, *Computing the Climate,* based on his in-depth study of multiple climate modeling institutions. Easterbrook writes "For example, Lenhard and Winsberg argue that design decisions built into these core elements [of GCMs] become deeply buried within the model over time – that is, *entrenched* – which means those decisions can no longer be understood or critiqued. But you only have to step into a climate modelling lab and talk to the modellers to realize this isn't the case at all" (2023, p. 142).

New sources mentioned:

Baumberger, Christoph, Reto Knutti, and Gertrude Hirsch Hadorn. 2017. "Building Confidence in Climate Model Projections: An Analysis of Inferences from Fit." WIREs Climate Change 8 (3): e454. https://doi.org/10.1002/wcc.454.

Carrier, Martin, and Johannes Lenhard. 2019. "Climate Models: How to Assess Their Reliability." International Studies in the Philosophy of Science 32 (2): 81–100. https://doi.org/10.1080/02698595.2019.1644722.

Easterbrook, Steve M. 2023. Computing the Climate: How We Know What We Know About Climate Change. Cambridge: Cambridge University Press. https://doi.org/10.1017/9781316459768.

Frigg, Roman, Erica Thompson, and Charlotte Werndl. 2015. "Philosophy of Climate Science Part II: Modelling Climate Change." Philosophy Compass 10 (12): 965–77. https://doi.org/10.1111/phc3.12297.

O'Loughlin, Ryan. 2023. "Diagnosing Errors in Climate Model Intercomparisons." European Journal for Philosophy of Science 13 (2): 20. https://doi.org/10.1007/s13194-023-00522-z.

Touzé-Peiffer, Ludovic, Anouk Barberousse, and Hervé Le Treut. 2020. "The Coupled Model Intercomparison Project: History, Uses, and Structural Effects on Climate Research." *WIREs Climate Change* 11 (4): e648. https://doi.org/10.1002/wcc.648.

1.2 The paper of Lenhard and Winsberg (2010) also demonstrates the failure of CMIP in making intercomparisons and thereby reaching what the authors call "component-level understanding". In this draft, the scientific references used to support the claim that AMIP / CMIP allowed for more component-level understanding are not recent (e.g. first paragraph p. 5 for instance or Glecker et al. 1995 cited p. 8), thus it would be nice that the authors explore whether it is the aim of CMIP6/7 using recent examples/illustrations. There might be another interesting paper on this topic, the paper of Touzé-Peiffer, Barberousse and Le Treut (2020).

**We see the evidence from AMIP/CMIP presented in Lenhard and Winsberg (2010) as only supporting a much weaker claim, i.e., that climate modelers did not achieve *as much* understanding of the sources of failures/successes in their models as they had hoped. Moreover, the reasons for this lesser degree of understanding are underdetermined—it could be due to model building/complexity (à la Lenhard and Winsberg), it could be due to the complexity of the model intercomparison effort itself (see O'Loughlin 2023), or some combination (or some other reasons). That's why we have not added further discussion of Lenhard and Winsberg beyond what was added in response to comment 1.1.  For a more thorough analysis of Lenhard and Winsberg 2010, see O'Loughlin 2023, pp. 3-14.**

**The manuscript does include discussions of some recent examples in addition to the older ones (e.g., see lines 235-243 in original (unrevised) manuscript). Our main aim is to show how, historically, CMIP has involved component-level understanding, error diagnosis, and fixing of models. Component-level understanding has been fruitful for CMIP. The explanation of progress made b/w IPCC AR3 and AR6 directly speaks to this. However, we agree that including another more recent example can strengthen the paper. To wit, we have added:**

In certain circumstances component-level responsibility can be determined. As an example, the Community Earth System Model 2 (CESM2) was recognized as exhibiting a too large climate sensitivity—one that did not appear in standard CMIP simulations. This behavior was discovered in a surprising way. Zhu et al. (2022) had shown an instability in the simulation of the last glacial maximum, a much colder period than present day, using CESM2. This instability did not exist in CESM. By reverting to the original, component-level microphysics scheme the model behaved as expected, and erroneous specification of microphysical particle concentrations were discovered and remedied. More generally, the understanding and observational constraint of ice microphysics is a challenge as demonstrated by the very large variations in ice water path across CMIP models. Using Perturbed Parameter Estimation (PPE, e.g., Eidhamer et al. 2024) can also reveal component level sensitivities and shortcomings.

**Sources:**

**Eidhammer, Trude, Andrew Gettelman, Katherine Thayer-Calder, Duncan Watson-Parris, Gregory Elsaesser, Hugh Morrison, Marcus van Lier-Walqui, Ci Song, and Daniel McCoy. 2024. "An Extensible Perturbed Parameter Ensemble (PPE) for the Community Atmosphere Model Version 6." EGUsphere, January, 1–27. https://doi.org/10.5194/egusphere-2023-2165.**

**J. Zhu, B. L. Otto-Bliesner, E. Brady, A. Gettelman, J. T. Bacmeister, R. B. Neale, C. J. Poulsen, J. K. Shaw, Z. McGraw, J. E. Kay, 2022: LGM paleoclimate constraints inform cloud parameterizations and equilibrium climate sensitivity in CESM2, *J. Adv. Model. Earth Syst.,* https://doi.org/10.1029/2021MS002776**

1.3 _ Another well-discussed issue in philosophy of science that makes "component-level understanding" difficulty to reach in the case of climate models is the epistemic opacity of climate models simulations, that models be dynamical or ML-based. Here are examples of such papers: Knüsel and Baumberger 2020; Kawamleh 2021; Jebeile, Lam and Räz 2021.

**We thank the reviewer for these paper suggestions.**

**First, we have added a discussion of epistemic opacity (see lines 368-375).**

**Second, we would like to point out here how our proposal differs from the analysis of Knüsel and Baumberger. Knüsel and Baumberger (2020) develop a framework to assess the fitness of climate models for providing understanding. They first show their framework applies to process-based dynamic modeling and then show that "data-driven models can be useful tools for understanding" as well. More specifically, along the dimensions of "representational accuracy" and "graspability," they demonstrate that a random forest model (a data-driven model) can satisfy these requirements as well.**

**However, our proposal differs from theirs. Take their "graspability" as an example. "Graspability" is defined as modelers' ability to qualitatively anticipate model outputs and their ability to explain model behavior. More specifically, random forest modelers can anticipate model outputs by "familiarizing themselves with the model through manipulation." They can explain model behavior by "studying the variable importance plot" "through manipulation" and "making inferences from the working of the optimization algorithm to model behavior." These ways of attaining graspability fall strictly within what post-hoc XAI methods can provide, which, in our paper, is statistical understanding. And we urge scientists to move beyond it. We now include in our manuscript (end of section 3) that "Overall, our analysis is in the same spirit as that of Knüsel and Baumberger (2020) who argue that data-driven models and dynamical models alike can be understood through manipulating the model so that modelers can qualitatively anticipate model behaviors. However, not all manipulations are equal. Manipulating input data and seeing associated**

**changes in output data does not tell you how the model produces its output.”**

**Third, as to Kawamleh (2021), we would like not to include it in our manuscript because the paper is built on a flawed belief about machine learning. The author attributes “the widespread failure in neural network generalizability to the lack of process representation.” Here, generalizability refers to the ML model’s ability to generalize beyond training data. More specifically, the author writes, “The ability to simulate climate change is a test of the generalizability of the NNP [neural network parameterization] beyond the training data.” But a naïve ML cannot generalize beyond training data, and *it is not supposed (or expected) to.* That is, while ML can generalize beyond the specific data it was trained on, it is not expected perform well on data that comes from entirely different distributions. One of us, Li (2023) explains (1) how precisely ML is automated induction; and (2) various problems of induction have counterparts in ML practice or theory; and (3) mitigation strategies in ML applications informed by philosophy. We see the Li (2023) paper as in line with Pacchetti, Jebeile, and Thompson (2024) who note that “The ability to train longer-time-scale ML models is lower… since we are always subject to the inductive problem that past performance does not guarantee future success.” For this reason, Kawamleh (2021) is both incorrect and irrelevant to our paper.**

**Finally, see our response to 2.1 below for a how our proposal relates to Jebeile, Lam and Raz (2021).**

**Source:**

**Li, Dan. 2023. “Machines Learn Better with Better Data Ontology: Lessons from Philosophy of Induction and Machine Learning Practice.” Minds and Machines, June. https://doi.org/10.1007/s11023-023-09639-9.**

2_ The authors put forward three kinds of understanding, instrumental understanding, statistical understanding and component-level understanding.

2.1 _ One would expect this taxonomy to be connected to what philosophers have already said about understanding with models, or to be motivated by what scientists tell about their own practices. Thus, in (Knüsel and Baumberger 2020) and (Jebeile, Lam and Räz 2021), the authors put forward different dimensions of understanding with models that therefore comes in degree. In particular, notably following the work of de Regt and Dieks cited in the paper, Jebeile, Lam and Räz (2021) use these evaluative criteria of understanding with models: intelligibility, representational accuracy, empirical accuracy, physical consistency, delimiting the domain of validity. Is this explicitation of “understanding with models” useful for this paper? For example, the difference between “statistical understanding” and “component-level understanding” is that only the latter meets intelligibility, no? (I am just curious here, this might not be crucial for the paper though).

**Indeed, Jebeile, Lam and Raz (2021) offer a comprehensive framework for model**

evaluation. While we agree with these criteria, not all concepts apply to the problem presented in our paper mainly because the problems that this framework is trying to address are different from ours. The problem we are trying to address is how we can build AI/ML models that can be diagnosed and improved in climate science—and we argue XAI doesn't really help as much as expected. This key question is suggested in our manuscript title: "Moving beyond post hoc XAI…" Hence our paper has a different focus than Jebeile, Lam and Raz (2021) do.

The distinction between statistical understanding and component-level understanding is introduced to serve our purpose of characterizing the limitations of post-hoc XAI methods in that XAI at best gives us the former, rather than the latter. Regarding whether intelligibility captures the difference between statistical understanding and component-level understanding, we think the answer is no. Jebeile, Lam and Raz (2021) define "intelligibility" as "the ability and skill of the agent to use the model and to obtain explanations from it, and on the features of the model that enable its manipulability." XAI methods also allow modelers to obtain explanations and manipulate the model. That is, both statistical and component-level understanding involve some degree of intelligibility. In this sense, the replacement of "component-level understanding" with "intelligibility" would collapse the distinction that we are trying to make. In our view, component-level understanding typically constitutes a higher degree of intelligibility than does statistical understanding, because agents will be better able to manipulate and generate explanations from a model when they can understand its innerworkings. As our examples demonstrate, this higher degree of intelligibility also involves diagnosing and correct model errors. Therefore, we would like to stick with statistical vs. component-level understanding.

However, thanks to your comment, we now see that our paper can offer a more nuanced version of intelligibility. The higher level of intelligibility maps on to manipulating model components; whereas the lower level to manipulating model input and output. We have revised the manuscript to reflect this (see lines 383-394).

2.2 _ What about the concept of "process understanding" used by climate scientists? It is usually referring to the aim of fundamental research. Is it not covered by the concept of "component-level understanding"?

They are related, often overlap in practice, but are distinct. "Process" refers to the physical process that the model aims to simulate. For example, Maloney et al. (2019) define process-oriented diagnostics as "characterizing a specific physical process or emergent behavior that is related to the ability to simulate an observed phenomenon." The concept is related to Jebeile, Lam, and Raz's (2021) "representational accuracy," which is "evaluated with regard to how well a model captures the relevant physical processes at work in the target system under investigation."

In contrast, having components is an engineering choice. For example, scientists could incorporate cloud formation by increasing the resolution of a GCM—increasing

the process representation (or what Knüsel and Baumberger call "representational depth"), and scientists can look at how clouds form in a GCM frame by frame to increase process understanding. But, in this case, cloud representation is not an isolated component in the model. It is merely a fine-grained small-scale process that also follows from the core physics. Conversely, scientists could build a cloud parameterization (a mini-model) that is connected to the original GCM. Now we have a component within the GCM. However, it is possible that the cloud parameterization offers very little process understanding whatsoever, as in the case of naïve or empirically based parameterizations.

**We've added a footnote to clarify:**

FN: Note that while processes and model components are linked, neither is reducible to the other. E.g., a coupler is a component in a GCM but it is not a real-world climate process; conversely, there is no cloud feedback parameterization but cloud feedbacks are a real-world climate process.

**\*See also lines 238-243 in our original submitted manuscript.**

2.3 _ In the paper, what is the role of this taxonomy after all? Couldn't the authors simply introduce the definition of "component-level understanding" (or process understanding) and argue that it can be reached in ML-based modeling with the help of XAI methods (where we could imagine that only statistical understanding is reached)?

**We have clarified the role of the taxonomy in the manuscript (see 388-390).. The key point is that our proposed hierarchy, in a sense, zooms in on differing degrees of intelligibility. Here is an excerpt from our revised introduction:**

In this Review and Perspective paper, we target readers with expertise in traditional approaches for climate science (e.g., development, evaluation, and application of traditional Earth System Models) who are starting to utilize ML in their research and who may see XAI as a tempting way to gain insight into model behavior and to build confidence. In this perspective, we draw from some ideas in philosophy of science to recommend that such researchers leverage the expanding array of freely available ML learning resources to move beyond post hoc XAI methods and aim for *component-level* understanding of ML models. By "component" we mean a functional unit of the model's architecture, such as a layer or layers in a neural net. By "understanding" we mean knowledge that could serve as a basis for an explanation about the model. We distinguish between three levels of understanding:

**Instrumental understanding:** knowing *that* the model performed well (or not); e.g., knowing its error rate on a given test.

**Statistical understanding:** being able to offer a reason why we should trust a given ML model by appealing to input-output mappings. These mappings can be retrieved by statistical techniques.

**Component-level understanding:** being able to point to specific model components or parts in the model architecture as the cause of erratic model behaviors or as the crucial reason why the model functions well.

These levels concern the degree to which complex models are intelligible or graspable to scientists (De Regt and Dieks 2005; Regt 2017; Knüsel and Baumberger 2020). Therefore, our proposal has a narrower but deeper focus than recent philosophy of science accounts of understanding climate phenomena *with* or *by using* ML and

dynamical climate models (Knüsel and Baumberger 2020; Jebeile, Lam, and Räz 2021). We are concerned with understanding, diagnosing, and improving model behavior to inform model development.

2.4 _ It would also be interesting to have a characterization of this taxonomy: is instrumental understanding a weaker form of understanding than statistical understanding? Is statistical understanding in turn a weaker form than component-level understanding?
Or do they overlap?

**No, they are not necessarily reducible to each other. For example, a technician might say, "I don't know why whenever you pat the TV really hard, it fixes the snowflakes on the screen, but I do know your TV needs a new set of LED backlight strips." Presumably, I, a layperson, discovered that patting on TV in a particular way will fix the snowflakes—statistical understanding—but this is unlikely to produce (in me) component-level understanding. Neither do statistical and component-level understanding overlap in an interesting way—except, maybe, in some lucky instances. They may overlap in a very general sense in that both include counterfactual probing (e.g., manipulate input data and see what happens vs. manipulate a model component and see what happens). However, statistical understanding *could lead to a discovery of* component-level understanding. Indeed, if the layperson decides to study patting and snowflakes, she may end up learning something useful and component-based about the TV. But it would be wrong to argue that these different levels can be reduced to each other.**

2.5 _ In the hierarchy of models envisioned by Held (2005), is he referring to model component?
In the quotations given p. 11, he instead speaks about the dynamics.
Another paper on hierarchy of climate models is (Katzav and Parker 2015).

**In the quote given, Held talks about understanding changes in model dynamics as model complexity increases by which he means, among other things, adding parameterizations. Further evidence of this can be found when he says: "My reading of the literature is that elegance is often sacrificed unnecessarily, primarily for the sake of competition with *comprehensive models*" (Held 2005 p. 1613, emphasis added). "Comprehensive" refers to representing more and more physical processes, which often involves adding more parameterizations.**

**We've added the citation to Katzav and Parker 2015.**

Regarding the contribution of this paper for the scientific practice:

1_ It would be worth defining what "computational efficiency" of machine learning is (introduction p. 2) as it is usually the main motivation in the use of machine learning. It would be important for this paper to clarify what it means.

**We have replaced it with "orders of magnitude reduced computational expense."**

2_ It seems that the authors are assuming (or have to assume) that there is some kind of isomorphism between layers in neural net and model components. Can they clarify their

position on this? (cf. second paragraph p. 4).

**Layers of a neural net do not necessarily need to be components. Components are just what scientists can isolate, turn on and off, *in* the model. If they choose to isolate layers or even neurons in a neural net, as shown in our examples, then these isolated parts serve as components. However, one can also imagine where mini neural networks serve as components and make up a big model.**

**For example, generalized additive neural net adopts a structure like this (see Agarwal, R., Melnick, L., Frosst, N., Zhang, X., Lengerich, B., Caruana, R. and Hinton, G.E., 2021. Neural additive models: Interpretable machine learning with neural nets. *Advances in neural information processing systems, 34,* pp.4699-4711).**

**Therefore, we do not have to assume layers must be components.**

3_ Can it not be that search for "component-level understanding" is actually search for "representational accuracy". Trying to correct for previous idealizations and parameterizations seem to be line with the "natural" direction of scientific research, no? (This is what is assumed in Jebeile and Roussos 2023; Baldissera Pacchetti, Jebeile and Thompson 2024).

**No, component-level understanding and representational accuracy are distinct aims. Representational accuracy refers to "how well a model captures the relevant physical processes at work in the target system under investigation" (Jebeile, Lam and Räz 2021). Representational accuracy implies a commitment to realism, as Pacchetti, Jebeile, and Thompson (2024) note, "… realism of the model assumptions. For a given system, there is a tendency to work toward more realistic representations, thus correcting for previous simplifications, notably through the integration of more variables and fine-grained details, or replacing previous parameterizations by explicit theory-based equations." As Pacchetti et al. have convincingly argued, the current trend of increasing resolution is motivated by the pursuit of representational accuracy.**

**Yet, representational accuracy is not always prioritized in AI/ML, at least not as much as it is in dynamic modeling. As Pacchetti, Jebeile, and Thompson (2024) themselves say, "Regarding realism, one characteristic of ML is that the data are not assumed to conform to given physical laws or regularities, instead relying on the emergence of those regularities in statistical form. In climate modeling, physical process representation has generally been the preferred form of modeling, with statistical or empirically derived parameterizations used only where physical processes are insufficiently well understood or would require a prohibitive share of the computing resource. Hard AI, or larger-scale implementation of ML methods to directly predict field-scale outputs, *is in conflict with the aim of realism*." (emphasis added.)**

**Furthermore, we argue that component-level understanding should be pursued mainly because it will serve scientists for diagnosis and improvement, rather than because it leads to realism. If realism is obtained, that's a bonus. Of course, in physics-informed ML, (see our example in section 4.2 of the manuscript) the goals of component-level understanding and representational accuracy may align, but they need not. To reiterate what we said in earlier responses, our aim is to focus on the component-level understanding *of* models rather than of real-world phenomena (although, of course, understanding and predicting real world phenomena is the ultimate goal).**

Lastly, we would like to stay away from a commitment to realism also because it implies a metaphysical commitment to a particular scientific worldview. For example, dynamic models use physics as core. Physical laws are expressed as partial different equations, which typically involves spatial derivatives $\partial x, \partial y, \partial z$, and temporal derivatives $\partial t$ of different physical quantities. These representations match the Newtonian worldview. Some may even argue that they match common sense. Yet, as Jebeile and Roussos (2023) have argued, "climate science retains much of its initial "physics-first" orientation, and that it adheres to a problematic notion of objectivity as freedom from value judgments." And the pursuit of representational accuracy based on physics limits methodological pluralism, as Pacchetti, Jebeile, and Thompson (2024) have argued. The same argument also applies if representational accuracy also includes "comprehensiveness" from Pacchetti, Jebeile, and Thompson (2024).

For these reasons, it is our deliberate choice to stay away from representational accuracy and simply focus on components that can aid model diagnosis and improvement, which, again is more about engineering practice than it is about realism.

4_As it is, section 4.3 fails to be persuasive because GAN does not to apply to climate science. The authors should explain whey they believe that, in the future, GAN will be applied to ML-driven climate science.

It turns out GAN is sometimes used in climate science (e.g., in Besombes et al. 2021 and in Berohe 2021).

It is also not unreasonable to think that GAN *dissect* could be used in future climate applications. Indeed, ablation (an XAI technique which involves turning on/off specific input data) has been used successfully in climate modeling already (e.g., see Brenowitiz et al. 2020).

We have revised the manuscript to reflect these points.

Sources:

Beroche, Hubert. 2021. "Generative Adversarial Networks for Climate Change Scenarios." URBAN AI (blog). April 2, 2021. https://urbanai.fr/generative-adversarial-networks-for-climate-change-scenarios/.

Besombes, Camille, Olivier Pannekoucke, Corentin Lapeyre, Benjamin Sanderson, and Olivier Thual. 2021. "Producing Realistic Climate Data with Generative Adversarial Networks." Nonlinear Processes in Geophysics 28 (3): 347–70. https://doi.org/10.5194/npg-28-347-2021.

Brenowitz, Noah D., Tom Beucler, Michael Pritchard, and Christopher S. Bretherton. 2020. "Interpreting and Stabilizing Machine-Learning Parametrizations of Convection." Journal of the Atmospheric Sciences 77 (12): 4357–75. https://doi.org/10.1175/JAS-D-20-0082.1.

Minor comments:

1_ In what sense, do AI models entail "greater uncertainty"? Could you specify what you mean: is that that the outputs / predictions of models are more uncertain? (abstract, p. 2)

**We don't claim that AI models "entail" greater uncertainty. In the abstract, we suggest that using black box models will subject climate science to greater uncertainty. The basic idea is that if you use a model that you do not understand, then the knowledge gained from the model could be more uncertain.**

2_ What is actually the "functional test" that a model has to pass in order to provide instrumental understanding? (abstract, p. 2)

**There are many possible ones. Basically, any performance assessment. What we have in mind are the types of performance assessments described, e.g., in lines 82-87 in the original submitted manuscript.**

3_ Some technical terms should be (better) defined: "layer-wise relevance propagation"; "attribution/relevance heatmaps"; "multi-layer, convolutional recurrent neural networks", "tree ensembles" (p. 6); distinction between "specific classificatory instances" and "global classification" (p. 7); "P-score" (p. 13)

**We will defer to the editor – should we explain these terms better?**

4_ There should be no bracket after "Gettelman et al. 2019." (p. 5).

**Thanks, fixed it.**

5_ References are needed to support the claim that "In addition, there is a concurrent need to establish the trustworthiness of ML models as driven climate science potentially becomes increasingly used to inform decision makers" (p. 5).

**Added this: "NSF AI Institute for Research on Trustworthy AI in Weather, Climate, and Coastal Oceanography (AI2ES)." n.d. Accessed August 13, 2024.** https://www.ai2es.org/.

6_ In the introduction of section 4, the authors write "we offer three examples in which ML researchers are able to acquire component-level understanding of model behaviors by intentionally designing or discovering model components that are interpretable and intelligible." This sentence seems to suggest that "interpretable and intelligible" model components will bring component-level understanding (p. 13). Could the authors clarify this point?

**We think that, in the examples we describe, having interpretable and intelligible model components exemplifies component-level understanding and helps scientists diagnose (and correct, if needed) model behavior.**

7_ The authors should write what the acronym PDEs is referring to in all letters (p. 14)

**Thanks -- fixed it**

8_ The authors should indicate the year of Pathak et. Al (p. 14) and add the reference in the list of references.

**Thanks – fixed it**

References indicated in this review:

Frisch, M. 2015. Predictivism and Old Evidence: a Critical Look at Climate Model Tuning. *European Journal for Philosophy of Science* 5 (2):171–190.

Lenhard, J., and Winsberg, E. 2010. "Holism, Entrenchment, and the Future of Climate Model Pluralism". *Studies in History and Philosophy of Science* Part B, 41(3):253–262.

Lenhard, J. (2018). Holism, or the erosion of modularity: a methodological challenge for validation. *Philosophy of Science, 85*(5) 832–844.

Jebeile, J., Lam, V. & Räz, T. (2021) Understanding climate change with statistical downscaling and machine learning. *Synthese* 199, 1877–1897. https://doi.org/10.1007/s11229-020-02865-z

Kawamleh, S. (2021). Can machines learn how clouds work? The epistemic implications of machine learning methods in climate science. *Philosophy of Science*, *88*(5).

Knüsel, B., & Baumberger, C. (2020). Understanding climate phenomena with data-driven models. *Studies in History and Philosophy of Science Part A*. https://doi.org/10.1016/j.shpsa.2020.08.003.

Baldissera Pacchetti, M., J. Jebeile, and E. Thompson, 2024: For a Pluralism of Climate Modelling Strategies. *Bull. Amer. Meteor. Soc.*, https://doi.org/10.1175/BAMS-D-23-0169.1, in press.

Touzé-Peiffer L, Barberousse A, Le Treut H. The Coupled Model Intercomparison Project: History, uses, and structural effects on climate research. *WIREs Clim Change*. 2020; 11:e648. https:// doi.org/10.1002/wcc.648

Katzav, J., & Parker, W. S. (2015). The future of climate modeling. Climatic Change, 132, 475–487.

**Citation**: https://doi.org/10.5194/egusphere-2023-2969-RC1

2. Reviewer 2 comments

Summary of paper:

This manuscript emphasizes the need to develop more interpretable AI models for climate applications, with emphasis on AI models that provide component-level understanding. It points

out that (posthoc) XAI methods that are applied after an AI model is built are not the way to go to achieve that goal. Instead, AI models should be built a priori to allow for component-level understanding. Comparisons are drawn to numerical climate models which tend to be built in components, making them easier to debug and interpret.

Comments:

I agree with the overall intent of the paper to push toward more interpretable AI models, rather than relying on XAI methods. While I agree with this intent, to me this seems to be a well-known goal and thus I do not see significant new contributions in this manuscript. Let me explain this section by section.

**While many climate scientists know about the importance of moving towards interpretable AI instead of relying on XAI methods, many do not. This may be especially true of our intended audience, climate modelers who are relatively new to ML.**

**To demonstrate this point, we collected publications related to AI, ML, or XAI, from three major journals (BAMS, GMD, JAMES). 178 valid references are acquired from the Web of Science. These references are categorized into one of the following based on their abstract: (1) Blackbox applications; (2) XAI applications; (3) Interpretable applications; (4) Review papers; and (5) Miscellaneous, e.g., workshop reports.**

**The results are:**

[Figure]

[Figure]

**The majority of publications still employ Blackbox AI methods, without explicit reference to interpretable components. There are roughly as many XAI papers as papers with interpretable models, which are still in the minority (around 10 papers each, in contrast to over 130 black box applications).**

**If deemed appropriate, we would be happy to include this data in the manuscript. We could look at more journals too, if desired.**

**Section 1** argues that relying on applying XAI methods after a model has been built has many drawbacks, and that Instead one should build models that are interpretable (what they call component-level understanding) from the start. However, this point has already been made many times. For example, the highly cited paper by Rudin (2019) (which is also cited in this manuscript) is entitled "Stop Explaining Black Box Models for High Stakes Decisions and Use Interpretable Models Instead," and makes this point very clearly: whenever possible build interpretable models, rather than relying on applying XAI methods after a model has been built. In the context of weather and climate the argument for interpretable models has been made many times, too, see for example:

- Yang, R., Hu, J., Li, Z., Mu, J., Yu, T., Xia, J., Li, X., Dasgupta, A. and Xiong, H., 2024. Interpretable Machine Learning for Weather and Climate Prediction: A Survey. *arXiv preprint arXiv:2403.18864*.
- Nhu, A.N. and Xie, Y., 2023, November. Towards Inherently Interpretable Deep Learning for Accelerating Scientific Discoveries in Climate Science. In *Proceedings of the 31st ACM International Conference on Advances in Geographic Information Systems* (pp. 1-2).
- Hilburn, K.A., 2023. Understanding spatial context in convolutional neural networks using explainable methods: Application to interpretable gremlin. Artificial Intelligence for the Earth Systems, 2(3), p.220093.

**We recognize that the call for interpretable models has been made in other work, and we thank the reviewer for pointing out some additional papers that make this point. This review/perspective paper makes a unique addition to the literature by demonstrating the parallels between a similar (and better known—to this community at least) challenge in building confidence in, and improving of, climate models. This unique addition is bolstered by the choice of GMD as the venue, as its audience consists of a mix of traditional and ML climate model developers and will therefore has better chances of reaching our target audience: traditional model developers who are starting to wade into ML and who may be tempted to rely solely on XAI for probing model behavior.**

**We have revised the introduction to make the contribution of this paper clearer.**

Section 2 outlines shortcomings and limitations of XAI methods. It mainly cites a few papers that have studied this topic. I did not find anything new here.

**We agree with this characterization of Section 2, though we are unsure whether the reviewer makes this point as a criticism. Given that the intended purpose of this paper is to help new ML model developers in the climate field to (a) understand XAI and its limitations, and (b) ultimately move beyond XAI, we argue that this outlining of shortcomings and limitations of XAI methods is necessary.**

Section 3 states that traditional (numerical) climate models tend to be based on components, which makes it easier to attribute problems to specific components of the model, and that the modularity of these components should be followed by AI models. Firstly, as one of the other reviewers already pointed out, the complex interactions of components in climate models can make it very difficult to attribute problems to individual components, so that reasoning does not always work for traditional climate models either. Secondly, traditional climate models are built in a modular structure because that is the only way humans can built such a complex system – by building one module at a time. Sure, that has other advantages as well – such as higher interpretability – but it wasn't the main reason. In contrast, modern AI tools are not naturally built on modularity, so it takes considerable effort to try to enforce modularity, especially for very complex tasks. Thus, in essence, I agree that it would be nice for AI models to be modular, but it might not always be possible.

**We break our response to this comment by the two separate points the reviewer makes in this paragraph.**

1.) "Firstly, as one of the other reviewers already pointed out, the complex interactions of components in climate models can make it very difficult to attribute problems to individual components, so that reasoning does not always work for traditional climate models either."

**This is a fair point, though "difficult to attribute problems" does not mean "impossible to attribute problems." We cite numerous sources and have added more in the revised draft, thereby demonstrating instances in which component-level understanding directly led to model improvements. We have also revised our language to be clearer that the issue of complexity may prevent component-level understanding from leading to useful insights. (See also our responses to reviewer 1, above).**

2.) "Secondly, traditional climate models are built in a modular structure because that is the only way humans can built such a complex system – by building one module at a time. Sure, that has other advantages as well – such as higher interpretability – but it wasn't the main reason. In contrast, modern AI tools are not naturally built on modularity, so it takes considerable effort to try to enforce modularity, especially for very complex tasks. Thus, in essence, I agree that it would be nice for AI models to be modular, but it might not always be possible."

**We are unsure whether it is fair to claim that "it wasn't the main reason," given that as far back as 1979, Schneider advocated for a hierarchy-based approach to climate model development that necessarily implies a modular design. Regardless, the original reasoning for a modular design approach, to climate models, is tangential to the point being made here: that modularity has had practical benefits for the understandability, interpretability, and ultimately the improvement of climate models. We are arguing that that practical benefit could be realized for ML models too.**

**Granted, the reviewer makes a fair point that current ML model design practices may make modularity more difficult. There is a value judgement in there: whether to value human time and effort versus valuing models that are, by design, amenable to component-level understanding and therefore interpretability and improvability. Making such a judgement call is beyond the scope of this paper; it is a judgement we would prefer to leave in the hands of climate researchers who are emerging as ML model developers.**

**Section 4** provides examples of three papers that are supposed to show how "component-level understanding" can be achieved for AI. How exactly is "component-level understanding" defined? Does it mean that we need to understand ONE component of the AI model? Or ALL components? If it's just one component – which it seems to be in several examples – then how is this different from physics-guided machine learning, which is an entire field? See for example:

- Willard, J., Jia, X., Xu, S., Steinbach, M. and Kumar, V., 2020. Integrating physics-based modeling with machine learning: A survey. *arXiv preprint arXiv:2003.04919*, *1*(1), pp.1-34.

**Note: we cite literature on physics-informed ML, so we are aware of this field. Here's a few things we can say:**

- **Understanding comes in degrees. Knowing one component gives some understanding; knowing more components gives more (see lines 138-140)**
- **Physics informed ML does not always involve component-level understanding (see section 4.1 of our manuscript)**
- **Component-level understanding is not limited to physics-informed ML (see section 4.2 of our manuscript)**

Also, there are many, many other examples from climate science that could have been cited, and I did not find any new ideas here.

**We added a co-author who is also an Earth System Model developer to help provide more perspective and relevant examples from climate science.**

**We also reiterate that the novel contribution of this paper is in the linking of existing climate model development practices to practices that could be employed in ML model development.**

**Section 5** recommends striving for component-level understanding. It's a good idea to strive for more modular AI architectures whenever possible, but I did not learn anything new here about how that could be achieved. I would also argue that we should focus on the more general topic of achieving interpretability, whether that is achieved through modularity or other means, such as feature engineering and/or symbolic regression.

**Review summary:** While I agree with the general idea that we should strive to make AI models more interpretable, unfortunately, I do not see any convincing new ideas here.

**Citation**: https://doi.org/10.5194/egusphere-2023-2969-RC2

**The reviewer's primary criticism in this last section seems to be "I did not learn anything new here about how that could be achieved" and "I do not see any convincing new ideas here." Those statements are understandable, given that the original draft was not clear in pointing out that (1) this is a perspective piece that aims to connect existing ideas in the literature and (2) the intended audience of this perspective is emerging ML model users and developers who may be tempted to rely on XAI methods. The reviewer is a well-known ML expert and model developer who is not in the intended audience for this perspective; it is understandable that the reviewer would not have learned anything new here. We have revised the paper to make clear that (a) this is a perspective piece, and (b) who the intended audience is.**

**3. Reviewer 3 comments**

**Summary:**

The authors hold the opinion that the artificial intelligence (AI) models should gain trust in the climate science community as the physics-based dynamical climate models do.

They proposed three types of understanding as a basis to evaluate trust in dynamical and AI models alike: instrumental understanding, statistical understanding and component-level understanding. The instrumental understanding is defined as knowing a model performed well (or not), or knowing its error rate on a given test. Statistical understanding is defined as being able to offer a reason why we should trust a given machine learning model by appealing to input-output mappings which can be retrieved by statistical techniques, and the component-level understanding refers to being able to point to specific model components or parts in the model architecture as the cause of erratic model behaviors or as the crucial reason why the model functions well. And they further argue that the currently Explainable artificial intelligence (XAI) models are only helping in increasing statistical understanding and hence not sufficient. They argue that the component understanding is essential for models to gain trust and propose for AI models to have interpretable components that are amenable to component-level understanding. Then the authors demonstrate some examples to support the arguments that XAI models only provide statistical understanding; dynamical climate models provide component understanding and finally AI models can (and should) have component understanding as well.

**Overall comment:**

The paper is clear and easy to understand with good writing. And it tries to address ML/AI model explainability which is a very important topic in climate science (or in any other areas), and argues that we should improve the explainability of AI models. However, I don't think this paper provides a comprehensive or innovative approach to achieve this goal. It seems to me that this paper proposes a concept that already exists in the common practice in the community. The key argument of the paper is to advocate for the 'component level understanding' which is essentially finding out which part of the model is not working, and tweaking or adjusting that part until it works. It is quite common for researchers to have some intuition or expectation on the functionality of each component in model architecture when they design a ML/AI model (including models applied to climate). Therefore the model naturally will have some 'component level' understanding albeit sometimes not explicitly decoupled. If the authors of the paper are arguing for the more explicitly decoupled or independent component for the model, I think they may need more concrete examples to illustrate the concept in the ML/AI models besides the current examples in the paper.

**Because this is a Review and Perspective paper, we do not claim to present an innovative new method. Our goal is to present a helpful way of thinking about understanding and explaining model behaviors. In particular, we draw the connection between diagnosing model behavior in traditional GCMs and in ML-driven climate science. Our paper is philosophical in character, as we now make clear in the revised introduction:**

In this Review and Perspective paper, we target readers with expertise in traditional approaches for climate science (e.g., development, evaluation, and application of traditional Earth System Models) who are starting to utilize ML in their research and who may see XAI as a tempting way to gain insight into model behavior and to build confidence. In this perspective, we draw from some ideas in philosophy of science to recommend that such researchers leverage the expanding array of freely available ML learning resources to move beyond post hoc XAI methods and aim for *component-level* understanding of ML models. By "component" we mean a functional unit of the model's architecture, such as a layer or layers in a neural net. By "understanding" we mean knowledge that could serve as a basis for an explanation about the model. We distinguish between three levels of understanding:

**Instrumental understanding:** knowing *that* the model performed well (or not); e.g., knowing its error rate on a given test.

**Statistical understanding:** being able to offer a reason why we should trust a given ML model by appealing to input-output mappings. These mappings can be retrieved by statistical techniques.

**Component-level understanding:** being able to point to specific model components or parts in the model architecture as the cause of erratic model behaviors or as the crucial reason why the model functions well.

These levels concern the degree to which complex models are intelligible or graspable to scientists (De Regt and Dieks 2005; Regt 2017; Knüsel and Baumberger 2020). Therefore, our proposal has a narrower but deeper focus than recent philosophy of science accounts of understanding climate phenomena *with* or *by using* ML and dynamical climate models (Knüsel and Baumberger 2020; Jebeile, Lam, and Räz 2021). We are concerned with understanding, diagnosing, and improving model behavior to inform model development.

**Session comment:**

In the following sections of the paper the authors take some examples to illustrate the three types of understandings. In section two the authors explain how an XAI method utilizes saliency map in convolutional neural networks to examine the input / output mapping and achieve statistical understanding, but they argue that XAI methods have limitation of not being able to distinguish between correlation and causation.

In session three the authors take other examples to argue that dynamical models, on the other hand, have component understanding. The examples include fixing errors in the Atmospheric Model Intercomparison Project by identifying and fixing ocean heat transport, and two more examples of updating parameterization help improve model performance and achieve component understanding. The example in this session is from 30 years ago and climate science has advanced a lot since then. It would be better to provide a more recent example.

**Thank you – we now include a more recent example (see lines 321-332).**

In session four the authors give three examples to claim that the AI models can achieve component understanding by either intentional model architecture design or finding interpretable model components. However, these examples are not persuasive enough to support the claims, especially for example three, which is in fact an ablation study, and XAI methods can also utilize this mechanism.

**We have revised the manuscript to highlight the relationship between the GAN dissect example and ablation studies (and we've added some citations; see lines 509 – 515). The key difference is that climate XAI applications of ablation typically involve turning input data on/off. As we argue in section 2 of the manuscript, this can only yield statistical understanding.**

**As for the persuasiveness of the examples overall, we reiterate that this is a Review and Perspective paper, not a presentation of novel research results. As we mentioned in response to reviewer 2, above, we realize that this point was not clear in the original manuscript and so we have revised to make our aim and intended audience clearer.**

In session five the authors further advocate for component level understanding and argue that the XAI methods can be complementary to the component level understanding.

Throughout the paper, the examples are briefly explained in plain text without detailed information or rigorous numbers to support their arguments. There are only two figures in the paper and they do not help much in explaining the contents in the text.

**Citation**: https://doi.org/10.5194/egusphere-2023-2969-RC3

---

## Author Response (AR2)

Authors' response to the editor

Thank you for the warm feedback and the suggestions. We have made all of the editor's suggested changes (details below). Please let us know if there's anything else we should do. We are looking forward to seeing our Review and Perspective paper published in GMD.

1) In response to reviewer #2, the authors presented results of a small study of ML publications from the journals BAMS, GMD, and JAMES. I think this is interesting and, because the authors mentioned that they would be willing to do so, I would like to encourage them to present this study in their manuscript. I note that this is optional, but I believe that presenting their statistics helps make clear the significance of their review and perspective paper. Please note that if this is added, the data should be made available and the data/code availability section updated to reflect this.

We have added this to the manuscript. In particular, we added a figure which shows the trends in ML publications at BAMS, GMD, and JAMES over time. We have uploaded the data and a description of the methods as supplementary material.

2) Reviewer #1 asked that several technical terms be better defined. The authors have asked for my judgement on the matter. Of the terms reviewer #1 mentioned, I agree that it would be good to better define or explain "attribution/relevance heatmaps", the distinction between "specific classificatory instances" and "global classification", and "P-score". Regarding that last item, I suggest that, where the authors have written "Indeed, NNU had a far lower P-score in both the baseline and the 4k warmer climate cases", they quantitatively state the reduction in P-scores. Reviewer #1 also mentioned "layerwise relevance propagation". I noticed that this is actually already explained in footnote #3, but the paper needs some reorganization to make this clear earlier, as the term is used in footnote #2 and the main body of the paper before footnote #3 is encountered. Reviewer #1 also mentioned "tree ensembles". I don't think this necessarily needs better explanation, but I might expand this to the more descriptive "ensembles of decision trees", and perhaps also note that a very common example off this is random forests. (I do not think that multi-layer convolutional neural networks needs explanation, as these are very commonly encountered in the literature in the context of geospatial data.)

We now define "attribution/relevance heatmaps" as suggested. We have moved our description of "layerwise relevance propagation" earlier, so readers will see it defined in a footnote on first use. We changed the "random decision trees" as suggested. We clarified

what is meant by "specific" vs "global" classificatory instances. Finally, we fixed the "P-score" description and corrected the relevant discussion (it was supposed to be a physical constraints penalty $P$, meaning that lower scores indicate less violation of physical constraints, given in units $W^2/m^4$). We've added quantitative values of $P$ to the manuscript as well.

Minor comment: In line 505 of the author-tracked changes, there is mention of a "certainty deficiency". I believe that the authors mean "certain deficiency"? (Or do they mean a deficiency in certainty? I am not entirely sure.)

Yes, it should be "certain deficiency". We fixed it – thanks!